



# Optimized Umkehr profile algorithm for ozone trend analyses.

Irina Petropavlovskikh[1,2], Koji Miyagawa[2], Audra McClure-Beegle[1,2], Bryan Johnson[2], Jeannette Wild[3,4], Susan Strahan[5,6], Krzysztof Wargan[6,7], Richard Querel[8], Lawrence Flynn[9], Eric Beach[10], Gerard Ancellet[11], and Sophie Godin-Beekmann[11]

[1]CIRES, University of Colorado, Boulder, 80306, USA
[2]NOAA GML, Boulder, 80305, USA
[3]CISESS, University of Maryland, College Park, MD, 20740, USA
[4]NOAA/NWS/NCEP/CPC, College Park, MD 20740, USA
[5]USRA, Columbia, MD 21046, USA
[6]NASA GSFC, Greenbelt, MD 20771, USA
[7]Science Systems and Applications, Inc., Lanham, MD 20706, USA
[8]The National Institute of Water and Atmospheric Research Ltd.: Lauder, NZ
[9]NOAA Center for Satellite Applications and Research, STAR, College Park, MD, United States
[10]IMSG, College Park, MD, United States
[11]Sevice d'Aeronomie, CNRS, France

*Correspondence to*: Irina Petropavlovskikh (irina.petro@noaa.gov)

**Abstract.** The long-term record of Umkehr measurements from four NOAA Dobson spectrophotometers was reprocessed after updates to the instrument calibration procedures. In addition, a new data quality-control tool was developed for the Dobson automation software (WinDobson). This paper presents a comparison of Dobson Umkehr ozone profiles from NOAA ozone network stations (Boulder, OHP, MLO, Lauder) against several satellite records, including Aura Microwave Limb Sounder (MLS; ver. 4.2), and combined SBUV and OMPS records (NASA AGG and NOAA COH). A subset of satellite data is selected
to match Dobson Umkehr observations at each station spatially (distance less than 200 km) and temporally (within 24 hours). Umkehr Averaging Kernels (AKs) are applied to vertically smooth all overpass satellite profiles prior to comparisons. The station Umkehr record consists of several instrumental records, which have different optical characterizations, and thus instrument-specific stray light contributes to the data processing errors and creates step changes in the record. This work evaluates the overall quality of Umkehr long-term measurements at NOAA ground-based stations and assesses the impact of
the instrumental changes on the stability of the Umkehr ozone profile record. This paper describes a method designed to correct biases and discontinuities in the retrieved Umkehr profile that originate from the Dobson calibration process, repair, or optical realignment of the instrument. The M2GMI and GMI CTM ozone profile model output matched to station location and date of observation is used to evaluate instrumental step changes in the Umkehr record. Homogenization of the Umkehr record and discussion of the apparent stray light error in retrieved ozone profiles are the focus of this paper. Homogenization of ground-
based records is of great importance for studies of long-term ozone trends and climate change.



## 1 Introduction.

The success of 30-years of international collaborations since the implementation of the Montreal Protocol and its amendments were celebrated at the Symposium for the 30th Anniversary of the Montreal Protocol (http://www.montreal30.io3c.org/) that brought together leading scientists, policymakers, and the public at the French Academy of Sciences in Paris, France (Godin-

Beekmann et al., 2018) on September 22-23, 2017. The emphases were on future scientific and public policy challenges for efficiently guiding ozone recovery processes (Newman et al, 2018). Confirmation of stratospheric ozone recovery was reported in recently published literature (Steinbrecht et al. 2017; Ball et al. 2019; SPARC/IO3C/GAW, 2019). The current state of stratospheric ozone recovery was summarized in the 2018 WMO/UNEP ozone assessment (WMO, 2018), where trend uncertainties for combined observational records have been used to describe confidence in detected trends. Uncertainty of

trend detection did not include full information about ozone measurement uncertainty. The difference in trends derived from satellite combined observational records suggests that further work needs to be done to assure good practices for the homogenization of long-term ozone records. Ground-based records are often used to verify the stability of satellite records (Fioletov et al, 2006, Krzycin and Rajewska-Wich, 2009, Nair et al, 2012, Flynn et al, 2014, Hubert et al, 2016, Bernet et al, 2019, Wang et al, 2020). In order to provide the reference, ground-based observations require careful and continuing

examination of past calibration records, changes in instrumentation and assessment of measurement uncertainties. Changes in the frequency of measurements can create complications in interpretation of relative stability of records and the resulting impact on the derived ozone trends (Sofieva et al, 2014; Damadeo et al., 2018).

Multiple studies show statistically significant positive trends in ozone in upper stratospheric levels in Tropical and Northern mid-latitudes, and nearly significant positive trends in the Southern Hemisphere. The statistical and analytical approaches to

quantify ozone recovery are complicated by the natural year-to-year variability which is detected in the observed ozone records. Moreover, stratospheric ozone recovery rates are expected to be slower than the decline of stratospheric ozone during the 1980s due to the long lifetime of the ozone-depleting substances. While ozone recovery in the upper stratosphere is mostly determined by halogen levels, temperature plays an important role in ozone recovery, including so-called "super recovery" , where ozone abundances exceed 1980 levels due to greenhouse gas-induced stratospheric cooling. At the same time, in the

lower stratosphere atmospheric composition and ozone levels are driven by the climate-impacted changes in the Brewer-Dobson circulation and by seasonal to decadal variability in stratosphere-troposphere exchange. These processes are difficult to discern and predict based solely on ozone or other atmospheric composition observations (Ball et al, 2019a; Ball et al, 2019b, Abalos et al. 2019; Orbe et al, 2020; Strahan et al, 2020; Dietmüller et al, 2021). Analyses of the processes that are responsible for ozone changes through atmospheric chemistry and dynamical transport rely on the development of Climate

Chemistry Models (CCM, Morgenstein, et al., 2018). However, the long-standing differences between the model reconstruction of the past ozone variability and observations suggest the need for improvement of simulations of the seasonal to sub-seasonal processes. Continuous verification of modelling results with the ongoing long-term measurements will help with understanding the processes that determine ozone recovery. Dobson Umkehr time series beginning in the 1950s are one



of a few long-term, historical ozone observational records. Continuous Umkehr datasets provide a reference for testing
consistency among shorter satellite and remote sensing methods and are used to validate combined records (Petropavlovskikh
et al, 2005; Kramarova et al, 2013).

The Umkehr method is based on measuring the difference in zenith sky intensities selected from two spectral regions (centred
on 311.5 and 332.4 nm, so-called C-pair) over a range of solar zenith angles (SZA). The longest records are those from Dobson
and Brewer spectrophotometers. The ratio of the observed radiances increases with increasing SZA and at about 86° SZA
reverses and starts to decrease up to 90° SZA, which grants the observation its name since Umkehr stands for reversal or
change in German. Using the Umkehr effect for calculating vertical ozone distribution was first described by Götz et al. (1934).
The earliest Umkehr measurements were performed in the 1930s at Arosa, Switzerland (Staehelin, 2017). The method helped
to determine the altitude of the maximum in the ozone layer and was applied around the world to study seasonal and interannual
cycles in ozone distribution. Several algorithms were developed to improve the Umkehr method and with an advance in
computers, the processing algorithm was developed by Mateer (1965). He investigated the impacts of the *a priori* and vertical
smoothing to assess the vertical resolution in the retrieved profile. The algorithm used Vigroux absorption cross-sections
(Vigroux, 1953). Carl Mateer applied his experience with the Umkehr method in developing the first algorithm for satellite
ozone retrieval (Mateer, 1971). After Bass and Paur published a new absorption cross-section and its temperature dependence,
Mateer and DeLuisi updated the Umkehr algorithm (Mateer and DeLuisi, 1992). DeLuisi et al. (1989) studied the effects of
volcanic aerosol interferences and found that stratospheric aerosols from the El Chichon eruption produced large errors in
Umkehr retrieved ozone profiles.

Despite short-term impacts from stratospheric aerosols on Umkehr ozone retrievals, the length and stability of the record were
considered as an advantage for satellite validation. DeLuisi (1996) provided reference to the SAGE I ozone data processing to
assist with the correction of its altitude registration. The analysis helped to homogenize SAGE I and SAGE II records for trend
analyses. Comparisons of Umkehr profiles with multiple SBUV(/2) ozone records (Bhartia et al, 2012) aided in assessment
of offsets between individual SBUV(/2) instrumental records due to satellite drifting orbit (Kramarova et al, 2013). Because
of its long-term measurement record, Umkehr data are regularly used for Scientific Assessments of Ozone Depletion (Harris,
Hudson and Phillips, 1998). They were first used in the early 1980s (Reinsel et al, 1984) to estimate changes in stratospheric
ozone depletion over long-term stations in US, India, Australia, Canada and Europe.

Some global locations that host a Dobson instrument have been providing routine, morning and afternoon, Umkehr
observations to the World Ozone and UV Radiation Center, WOUDC, database, including a number of stations hosting
Umkehr time series that start in the late 1950s. This renders the Umkehr ozone profiles the longest ozone profile time series
(Bojkov et al., 2002) and is central in validating other observational methods (Petropavlovskikh et al., 2005a), as well as
numerical models that simulate and forecast ozone content changes (Zanis et al., 2006). These profiles do not come as a
replacement to other ground-based observations of the ozone profile, but rather serve to complement them. Ozonesonde
observations provide a much finer vertical resolution profile; however, the measurements typically stop at the balloon burst
altitude of 30 km (Deshler et al., 2008), and the launches are typically performed once a week or less (with the exception of



two stations in Europe where sampling is done three times a week). The issue of relatively short time records also applies to Lidar (Jiang et al., 2007) and Microwave (Moreira et al., 2017) observations.

The Umkehr retrieval algorithm relies on the "self-calibration" technique that applies normalization of a set of morning or afternoon measurements to a single measurement selected at the smallest SZA. This process removes the majority of the instrumental artifacts and homogenizes time series. The vertical distribution of ozone is retrieved in 10 ozone layers between surface and ~45km.  However, routine (operational) data processing is still not optimized to account for an out-of-band (i.e. known as stray) light that affects measurements at the high SZAs (Petropavlovskikh et al, 2005b; Evans et al., 2009).

Optimization of stray light correction is a unique process to each Dobson instrument as it depends on its band-pass and optical alignment that are not always known from the historical calibration records. Recent attempts to measure the band-passes of several Dobson instruments in the optical lab with lasers (Kohler et al., 2018) led to an investigation of instrumental uncertainties in Dobson total ozone retrieval. The band-pass adjustment for some instruments lead to several percent change in derived total column ozone. However, not many instruments have been optically characterised so far. The Dobson Umkehr

algorithm thus requires an extensive verification of stray light levels in multiple instruments used to create long-term records. Change of the instrument can introduce step changes in the vertical distribution of retrieved ozone profiles and thus affect the stability of the long-term record.

NOAA Dobson ozone observations are positioned to continue monitoring stratospheric ozone recovery for the next 30 years. In addition to the six NOAA Dobson Stations (Table 1) and four NOAA Brewer stations), Umkehr observations are regularly

performed by several Dobson (3) and Brewer spectrometers (6) that are distributed globally. Stratospheric ozone recovery rates will differ between tropics, middle latitudes and high latitudes (WMO/UNEP Ozone Assessment, 2018). Umkehr stations are located at multiple locations around the world and will hence provide important information for tracking ozone recovery. The current operational Umkehr profile algorithm produces data that have relatively large uncertainty (~ 5 % in the stratosphere), which precludes our ability to detect small changes in stratospheric ozone. The refinement of the processing

software is required to resolve the instrument-related offsets in ozone profile retrievals. It is also important to remove offsets between satellite and ground-based ozone profiles to further improve the satellite ozone profile validation process. The main objective is to add value to the validation activities with the continuous improvement of the satellite retrieval algorithms that require new ground-based observations of higher accuracy while additionally reducing the noise in the data improving the usefulness for trend analysis.

In this paper we discuss optimization approach to homogenize long-term Umkehr ozone profile records. In Section 2 we describe several long-term ozone observing records and model simulations of stratospheric ozone variability selected for this study. We also discuss a matching criterion for comparisons of these records with ground-based observations.  In Section 3 we present methods developed for identification of vertical and temporal offsets between operational Umkehr and other ozone observing systems. Then, we describe the approach for removing offsets to homogenize Umkehr record. Finally, in Section 4,

we demonstrate the consistency between optimized Umkehr and other ozone records.



## 2 Data.

### 2.1 NOAA Dobson total ozone and Umkehr ozone profiles.

Dobson total column ozone records are regularly used in satellite record validation (Bai, 2015; Koukoulil, 2016; Boynard, 2018) and development of the global combined ozone data records (Fioletov, 2008; Hassler 2018). In 2017 NOAA long-term

Dobson total column ozone records at 15 stations were homogenized to account for inconsistencies in the past calibration records, data processing methods and selection of representative data. The updated total ozone records are used in Umkehr ozone profile retrievals. Descriptions of three Dobson stations used in this paper analyses, instrumentation, and total ozone data changes can be found in Evans et al. (2017) paper.

The ozone profile data at NOAA are collected by Dobson instruments with Umkehr method deployed only at 6 ground-based

stations: Fairbanks, AK, US; OHP, France; Boulder, CO, US; Mauna Loa, Hawaii, US; Perth, Australia; and Lauder, New Zealand (Table 1). Observations at all Umkehr stations are ongoing except at Perth where the Dobson stopped collecting data in 2016. In this paper we focus our discussion of changes in Umkehr record at Boulder, CO. The Appendix A shows summary results for OHP (middle northern latitude), MLO (tropical latitude) and Lauder (South Hemisphere middle latitude) stations, while results for Fairbanks (high northern latitude) and Perth (middle southern latitude) are similar to other stations and

therefore are not shown.

The Umkehr data collection is automated by the NOAA WinDobson operational software (Evans et al., 2017) that schedules zenith sky observations at C-pair spectral channels during the morning and afternoon hours. The software uses the near-IR cloud detector to screen the Umkehr data for clear sky conditions, interpolates screened observations to 12 nominal SZAs, adds total column ozone information, processes data and checks retrieved ozone profiles for quality flags and against station

climatological variability (+/-2 standard deviations).  NOAA Dobson Umkehr operational ozone profile data are posted on the GML archive https://gml.noaa.gov/aftp/data/ozwv/Dobson/AC4/Umkehr/. The Umkehr observations are archived at the WMO ozone and UV Data center (www.woudc.org), operated by the Environment Climate Change Canada, where the centralized data processing is done by python-based version of the UMK04 processing software (https://github.com/woudc/woudc-umkehr). The content of the files at the NOAA and WOUDC archives is the same for the

operational Umkehr ozone profile record, but the format differs.

### 2.2 Ozonesonde data.

The ozonesonde instrument has been launched on the meteorological balloons since the 1980s at ten NOAA stations. Evolving instrumentation has created discontinuities and gaps leading to inhomogeneous data records. NOAA and the international community developed homogenization methods for NOAA and SHADOZ networks (Sterling et al, 2018; Witte, 2018). The

error budget for each profile is calculated and included in the archived files (Sterling, 2018). Modern ozonesonde instruments sample ozone at the high vertical resolution, on the order of 100 – 200 m. The sondes constitute an essential component of satellite calibration and cross-calibration (Hubert, 2016), and are used for verification and improvement of climate chemistry,





chemistry-transport models and reanalyses (Stone et al, 2016; Miyazaki and Bowman, 2017; Wargan, 2018; Stauffer, 2018). The ozonesonde profile records provide key measurements for the middle and lower stratospheric, and tropospheric ozone

trend calculations, and are a benchmark network for stratospheric ozone profile observations (Steinbrecht, 2017; SPARC/IOC/GAW, 2018; WMO, 2018). Data for ozonesonde records are publicly available from the NOAA Global Monitoring Lab (GML) at https://gml.noaa.gov/aftp/ozwv/Ozonesonde/, from the World Ozone and Ultraviolet Radiation Data Centre (WOUDC) at www.woudc.org, from the Network for the Detection of Atmospheric Composition Change (NDACC) at www.ndacc.org, and from the NOAA National Centre for Environmental Information (NCEI) archive at

https://data.nodc.noaa.gov/cgi-bin/iso?id=gov.noaa.ncdc:C01562. In this paper we are using ozonesonde data from Boulder, USA; Hilo, USA; and Lauder, New Zealand. The data for the first two stations are taken from the NOAA GML archive and are homogenized version (Sterling et al, 2018). The Lauder ozonesonde data prior to 2018 were provided by Richard Querel of NIWA, New Zealand for the use in the LOTUS Report (2019). This dataset is not homogenized, and the data are the same as archived at NDACC (http://www.ndaccdemo.org/). We extended Lauder ozonesonde data with the un-homogenized 2018-

2020 data downloaded from the NDACC archive (last accessed in April 2021). The OHP ozonesonde data were homogenized in 2020. The data are available from the NDACC archive (Gaudel et al., 2015). However, the NDACC version at the time of data analyses contained some small errors associated with the telemetry noise in the recent measurement period. Therefore, we used the latest version provided by G. Ancellet and S. Godin-Beekmann of Latmos, France (private communications, June 15, 2021), which is also now archived at NDACC.

**2.3 Satellite ozone profile data**

Several satellite records are used for monitoring ozone globally and vertically. In this paper we are using daily NOAA and NASA long-term records that are sampled for the Umkehr station overpass conditions and also matched in time with Umkehr profiles.

**2.3.1 SBUV and OMPS ozone profile records**

NASA and NOAA have produced satellite measurements of ozone profiles through the Solar Backscatter Ultraviolet (SBUV) and related instruments (Nimbus 4 and 7) providing nearly 40 years of continuous data (1978 - present). The use of the common-design single instrument dataset eliminates many homogeneity issues including varying vertical resolution or instrumentation differences. Version 8.6 SBUV data incorporates additional calibration adjustments beyond the Version 8 release (McPeters, 2013, Bhartia et al, 2012). Small but evident biases remain (Kramarova, 2013).

The Suomi National Polar-orbiting Partnership (S-NPP) satellite of the Joint Polar Satellite System (JPSS) was launched in October 2011 (Flynn et al, 2006). It carries the Ozone Mapping and Profiler Suite Nadir Profiler (further referred to as OMPS) sensor that collects high spectrally resolved solar backscattered radiance in the sun-lit part of the globe (Seftor et al, 2014). OMPS makes measurements from 250 to 310 nm with a 1.1 nm resolution. It has a 16.6° cross-track FOV and 0.26° along-track slit width, but several spectrums are combined to cover a footprint of 250x250 km. The ozone profile retrieval is very



similar to Rodger's optimal statistical method deployed in the SBUV and Umkehr retrieval techniques. Validation of the NOAA operational OMPS ozone profile products is described in Flynn et al. (2014). Evaluation of the OMPS NASA V8.6 algorithm products for trend analyses is described in McPeters et al. (2019).

In this paper we used two satellite combined records. The first record is the NASA aggregated dataset (further referred to as AGG) which is comprised of SBUV, SBUV/2 and OMPS profiles from all (Nimbus 4 through NOAA 19) overlapping satellites

and using the NASA version 8.6 processing (McPeters et al. 2013). The AGG station overpass data are selected from all daily records that are found within the +/- 2/20 degrees latitude/longitude box centred on the station location and averaged using 1/distance weighting to the station location. The data set for Boulder station is available at https://acd-ext.gsfc.nasa.gov/anonftp/toms/sbuv/AGGREGATED/sbuv_aggregated_boulder.co_067.txt. The AGG overpass records for other Umkehr stations can be found in the same directory. Sometimes, there are 2 or 3 satellite overpass data found for a single

day. For the purpose of comparisons with Umkehr data all daily records are averaged.

The second record is the COH data set that combines records data from the SBUV/2 and OMPS (NOAA processing, further referred to as OMPS_NOAA) instruments on the many satellites using correlation-based adjustments providing an overall bias adjustment plus an ozone dependent factor (SPARC/IO3C/GAW. 2019). The resulting profile product is a set of daily or monthly zonal means, has been used in climate reviews (Weber, 2018; Steinbrecht, 2017) and is publicly available at https:

ftp.cpc.ncep.noaa.gov/SBUV_CDR.

In order to create the station overpass data each SBUV/2 and OMPS satellite record is sampled separately to find all daily records from +/- 2/20-degree latitude/longitude box centred on the station, The collected profiles are 1/distance weighted to the station location and averaged. This is a similar process to the AGG overpass record but does not combine daily data from different satellites. The overpass data from each satellite is adjusted using the SBUV COH technique developed for zonal

average data. The SBUV/2 & OMPS COH station overpass data (further referred to as COH) are available at NOAA website at https://ftp.cpc.ncep.noaa.gov/SBUV_CDR/overpass.

### 2.3.2 Aura MLS profiles

The Microwave Limb Sounder (MLS) measured ozone profiles from the UARS and Aura satellite platforms (Waters et al, 1999). We use Aura MLS Version 4.2 data (Livesey et al, 2020) for comparisons with Umkehr observations during the 2005

– 2020 period. MLS Version 5.1 was not available at the time of analysis, the ozone product is not expected to differ significantly between the two versions (Levesey et al, 2020). Ozone profiles are provided on 12 pressure levels per decade, the vertical resolution of MLS AK is about 2.6 km in the middle stratosphere and increases to ~3.5 km at 1 hPa pressure level. The MLS mixing ratio profiles are converted to layers in DU using pressure and temperature profiles provided in the files as also measured by MLS. The Umkehr AKs are applied to smooth MLS gridded profiles prior to comparisons. The frequency

of MLS observations in space and time (3500 profiles daily between 82-degrees N and 82 degrees S latitudes) provides matching overpasses within ±5° latitude and ±5° longitude of the Umkehr station location. Validation of the accuracy of MLS ozone profiles and their stability is described in Livesey et al. (2020). The MLS ozone profiles are assimilated in the MERRA-





2 reanalyses (Wargan et al, 2017 and references therein). Section 2.4 discusses MERRA2 data use in the global NASA chemistry transport models used for Umkehr homogenization.

### 2.3.3 SAGE II ozone record

SAGE is an ongoing series of solar occultation instruments spanning several decades providing high-precision vertical profiles of ozone from the troposphere to the mesosphere with ~1 km vertical resolution. Providing the longest single-instrument record of stratospheric ozone, SAGE II (Mauldin et al., 1985) was operational onboard the Earth Radiation Budget Satellite between October 1984 and August 2005. In this paper we use the 1985 and 2000 period to avoid the reduced sampling after 2000. In mid-inclination orbit (57°), the instrument observed upwards of 31 solar occultation measurements per day (~15 sunrises and ~15 sunsets as viewed from orbit). The sampling is such that, for each event type, successive observations are evenly spaced in longitude (i.e., ~24° between each) and slowly moving in latitude, collectively providing uniform sampling over two separate latitude bands of different meridional extents (i.e., larger near the tropics and narrower at mid-latitudes) in any given day that slowly shifts from day to day. Because of the infrequent sampling, the matching criteria for the SAGE II ozone satellite data is relaxed to +/- 20 degrees in longitude and +/- 2 degrees in latitude. The SAGE II ozone V7 data are available as number density profile at pressure levels from this directory:https://doi.org/10.5067/ERBS/SAGEII/SOLAR_BINARY_L2-V7.0 . The number density profile is converted to ozone partial pressure and to DU (1 DU is $2.69 \times 10^{20}$ molecules per meter squared) using pressure and temperature profiles provided in the files which are based on MERRA. The high-resolution SAGE II profile is smoothed with AK from the respective Umkehr profile found by temporal and spatial matching as described above.

## 2.4 GMI CTM and M2GMI simulated ozone profiles

The NASA Global Modeling Initiative chemistry transport model (GMI CTM), an off-line model driven by MERRA2 meteorological reanalysis (Gelaro et al., 2017), is used to assess the impact of various natural and anthropogenic perturbations of atmospheric composition and chemistry (Strahan, 2013). Strahan et al. (2016) uses the excellent agreement between simulated and observed seasonal evolution of Arctic $N_2O$ to demonstrate the simulation's value in quantitatively separating chemical from dynamical changes in polar ozone depletion during the Aura period (2004-2015). Douglass et al. (2017) compared a GMI CTM simulation with mid-latitude NDACC column measurements of long-lived reservoir species $HNO_3$ and HCl to verify the realism of MERRA2 transport in both hemispheres from 2004 to the present and to demonstrate the value of GMI CTM simulations to explain how sparse sampling impacts interpretation of trends in the observations. Strahan et al. (2015) analysed MLS $N_2O$ data to show that the QBO had a profound and far-reaching impact on $Cl_y$ variability in the Southern Hemisphere. The QBO modulates the extratropical mean age (and hence $N_2O$ and $Cl_y$) each winter, and the impacts are then transported to the Antarctic lower stratosphere on a one-year time scale. The QBO adds unexpected interannual variability to Equivalent Effective Stratospheric Chlorine (EESC) in the southern extratropical stratosphere.





The CTM is integrated at 1-degree horizontal resolution on 72 vertical levels from the surface to 0.01 hPa and uses MERRA2
meteorological fields as input. The output from the GMI CTM simulation is available for 1985-present
(https://portal.nccs.nasa.gov/datashare/dirac/gmidata2/users/mrdamon/Hindcast-Family/HindcastMR2V2/).    The    CTM's
tropospheric physical processes include convection, boundary layer turbulent transport, wet scavenging in convective updrafts,
wet and dry deposition, lightning $NO_X$ production, and anthropogenic, natural and biogenic emissions. The chemical
mechanism uses JPL-2015 rates and currently has 119 species and more than 400 kinetic and photolytic reactions; it is an
updated version of the mechanism described in Duncan (2007).

Customized  GMI CTM simulation outputs were created for the three NOAA Dobson Umkehr stations for 1979-2017 to assist
in the assessment of the instrumental offsets and to develop instrument-specific corrections to homogenize Umkehr record.
GMI CTM data at the NDACC sites (including six NOAA Umkehr sites) is available at www.ndacc.org. The files contain
vertical profiles of O3, NO2, H2O, temperature, pressure, potential temperature, and potential vorticity on a geometric altitude
grid with hourly time resolution. Model output is generated on geometric altitude, geopotential height, or pressure level grids
as needed for comparisons with Umkehr that is derived as pressure level gridded layer data. Daily global ozone, trace gas, and
meteorological fields are also available as needed for synoptic-scale interpretation of Dobson and ozonesonde data.

We use another simulation M2GMI (Orbe et al, 2017, Wargan et al, 2018) that is available for Umkehr step-change analyses.
It is called MERRA-2 GMI ("M2GMI"). M2GMI is the full GEOS general circulation model (GCM) with the GMI chemical
mechanism and is driven by the MERRA-2 horizontal winds, temperature, and surface pressure using the 'replay' methodology
(Orbe et al., 2017). The MERRA-2 assimilated meteorological fields are used by the model to simulate meteorology that is
continuously adjusted to the MERRA-2 winds, temperature and surface pressure. Comparisons of the M2GMI against
MERRA2, GMI CTM, and ozonesonde profiles have been recently described in Stauffer et al (2019).

The step-change in the GMI CTM ozone record in 1998 was documented (Stauffer et al, 2019 and references therein). The
TOVS/ATOVS satellite data transition in 1998 and assimilation of MLS temperatures in 2004 impacted the MERRA-2
meteorological fields (Gelaro et al., 2017). The MERRA2 analysis increments alter the wind fields that come from its general
circulation model (GCM), pushing them toward the meteorological observations. Where the GCM has biases, the increments
are large, driving unrealistic circulations that impact the GMI CTM stratospheric ozone distributions in the tropics and
subtropics.

There are differences between the GMI CTM and M2GMI ozone simulations. Even though they both use the same full GMI
chemical mechanism, the meteorology used in the 2 models is not identical. In the GMI CTM the MERRA-2 meteorological
product is used. M2GMI output is driven by a specified dynamics (SD) simulation. Instead of using MERRA-2 meteorology,
this SD uses a different method: "replay" (see further description in Orbe et al, 2017). Because the 1998 and 2005 discontinuity
is smoothed in the M2GMI ozone record (Stauffer et al, 2019), we decided to use its ozone data as a reference for the Umkehr
optimization. In addition, we are using the GMI CTM output for assessment of changes in the optimized Umkehr record and
for evaluation of ozone variability represented by two modelling records.



The M2GMI ozone profile output is sub-sampled for Boulder, OHP, MLO (or Hilo) and Lauder Dobson station geolocation (selected from the grid closest to the station location) and is matched within 30 minutes to the Umkehr observation (local time for the averaged sun elevation between 70 and 90 degrees SZA). The ozone profiles are provided on the constant pressure

levels that are converted to DUs and smoothed with Umkehr AK to created Umkehr-like layers. This is the version of data that is used as a reference dataset for Umkehr optimization. The M2GMI ozone and temperature profiles are available for 1980-2019 time period (https://www.esrl.noaa.gov/gmd/aftp/data/ozwv/Dobson/AC4/). In addition, the temperatures are used to adjust ozone absorption cross-sections in the radiative transfer modelling of Umkehr curves to account for the diurnal, daily and seasonal ozone variability in stratosphere (See Appendix D).

## 2.5 FG11 and QBO a priori


FG 11 (further referred to as fg11ap) is a climatological ozone dataset (McPeters and Labow, 2011) that describes typical ozone variability with latitude (5 degree zonal averages) and season (12 months). It is based on the Aura MLS and ozonesonde records measured between 2005 and 2010. Note that the ozone profile on any day of the year is the same in each year of the record. Thus, ozone in each Umkehr layer only changes seasonally.

The QBO a priori (further referred to QBOap) is an ozone climatology developed for analyses of the SBUV records to improve soft calibrations for the MOD ozone record (Ziemke et al., 2021). In addition to the seasonally and latitudinally dependent climatology the method empirically modifies ozone profiles based on the phase of the QBO cycle. The QBOap is a zonally (36 5-degree latitude bins) and monthly averaged dataset available from 1970 to 2019.

Both climatologies are matched with the dates and latitude location of the Umkehr observation at Boulder stations (40.05 N)

and are also AK -smoothed.

## 2.6 Combined MLS and ozonesonde record

The Aura MLS record (described in section 2.3.2 above) is matched with ozonesonde profile by date (+/-12 hours) and location (+/-5 degrees in longitude and +/- 5 degrees in latitude). The approach to the combining of MLS and ozonesonde record is described in McPeter and Labow (2011). We use this method to extend MLS station overpass ozone profile below 100 hPa

with the ozonesonde profiles. The time series of MLS-ozonesonde combined profiles between 2005 and 2020 is created for Boulder station. The extended dataset is indicated by SND_MLS in the figures and is used in the homogenization process.

## 3. Optimization of Umkehr stray light corrections.

## 3.1 Description of the Dobson measurement uncertainties.

The Dobson consists of two monochromators and a slit plate for selecting two bands of the UV solar spectrum approximately

20 nm apart. An optical wedge and photomultiplier tube are used to determine the relative intensity of the pair (see Komhyr and Evans, 2006 for further details). It has been demonstrated (i.e. Moeni et al, 2019) that each Dobson instrument has a





unique optical system. Some of the optical wedges are made from fused silica and others from quartz glass. Fused silica has higher UV transmission and is relatively even across the spectra used by the Dobson. The transmission of quartz glass is several percent less and passes longer wavelengths more efficiently. The optical wedges are also designed to have a logarithmic density

curve, but wedge calibrations show that it's not uniform across the entire wedge, and some are inherently darker overall. An error in poorly mapped wedge tend to increase toward the darker portion of the wedge, which would have a greater effect on measurements made at large SZAs. The thickness of the cobalt filters can make observations at longer wavelength more susceptible to stray light.

With time, the optical alignment in the instrument may shift or the optical prisms may degrade. These changes are identified

during the calibration procedures (every 4-6 years) and post-corrected to homogenize the ozone record at the station. The total ozone changes are typically corrected with a linear adjustment (step change or time dependent increments based on comparison with the Dobson standard), but for Umkehr measurements the changes are identified through the characterization of the optical wedge which is then mapped into R-N tables that produce Umkehr N-values. The relation between R and N are not linear and thus can modify the shape of the Umkehr curve after the calibrations. This is a small change in N-value but can result in a

significant (above uncertainty) step change in the Umkehr ozone profile.

The optical characterization of instruments is not a simple task. It often requires mapping of the Dobson response to the laser beam that shines on the Dobson optics (Christodoulakis et al, 2015; Evans et al, 2009). The original method was developed by Dobson (1957), but it is now done with two standard lamps (Komhyr and Evans, 2006). The calibration N tables are changed when the difference between the station and reference instrument is greater than the equivalent of 1 % in TOC.

Recent investigations of the difference in the bandpasses of three reference Dobsons (regional standard Dobsons No. 064, Germany,No. 074, Czech Republic, and the world standard No. 083, USA) were performed in a laboratory setting with support from the EMRP ENV 059 project "Traceability for atmospheric total column ozone" (Kohler et al, 2018). Although some small deviations in the band-passes were found, the effective absorption cross sections derived using each Dobson slit function did not differ significantly and thus affected the derived total column ozone by less than 2 % (depending on the ozone cross

section and wavelength pair). Unfortunately, the laboratory setting did not allow assessment of the stray light contribution for the three Dobson instruments.

The non-laboratory-based methods can be used to discern the level of the stray light when referenced against another instrument with similar (Christodoulakis et al, 2015) or higher level of stray light rejection (Moieni et al, 2019). However, even with the knowledge of the instrument specific band-pass (shape and spectral alignment) and with the expected level of

stray light (between 10^-4 to 10^-5) a small, but significant SZAs dependent bias remains unexplained in Umkehr observations. Moreover, this bias propagates into the retrieved Umkehr profiles and creates a 5-10 % bias relative to other ozone observing techniques (Petropavlovskikh et al, 2011). The next session demonstrates the standardised stray light corrections and changes in the Umkehr biases.





## 3.2 Standardised stray light corrections.

The impact of a stray-light induced error in the Umkehr retrieval is described in Petropavlovskikh et al. (2009) where Umkehr profiles in Boulder were compared against NOAA-11 and NOAA-16 SBUV/2 V8 satellite and ozonesonde co-incident profiles. It is further demonstrated in this paper by comparing multi-year biases between operational Umkehr retrievals at three additional stations (Haute Provence, France, Mauna Loa, Hawaii and Lauder, New Zealand, see Table 1 for details) and several satellite records (Aura MLS, AGG and COH, see details in Table 3). Prior to comparisons, all records with vertical resolution

less than 2 km (satellites and ozonesondes) are converted to DU, interpolated to 61 pressure levels (quarter of a standard Umkehr pressure layer) and smoothed with the Umkehr AKs. Subsequently, the high-resolution profiles are integrated to the ten standard Umkehr layers (see Table C1). Figure 1 shows comparisons for Umkehr profiles at Boulder, CO processed with (a) operational retrieval and (b) with application of standardized correction for stray light. Similar plots for Umkehr records at OHP, MLO and Lauder appear in Appendix A.

Two panels in Fig. 1,a summarize biases for operational Umkehr profiles. Ozonesonde profiles are matched between observations and models in time of Umkehr observations in Boulder (+/-12 hours) and space (+/- 50 km). Biases in 8 Umkehr layers are averaged in two periods (before and after 2005). The left panel shows comparisons between Umkehr and GMI CTM, M2GMI, AGG, COH and ozonesondes. The right panel also includes comparisons with Aura MLS. AGG and COH results are nearly identical supporting the consistency of the two different combination tactics. The COH bias does not change

significantly before and after 2005, it agrees well with operational Umkehr in layers 2, 5 and 6, while it shows higher ozone in other layers with the largest positive bias (up to 15 %) in layer 8. The bias in layers 3, 6, 7 and 8 are larger than 5 % that is Umkehr retrieval uncertainty for these layers. Layer 1 bias is also larger than 5 %, but Umkehr retrievals uncertainty in this layer is ~ 10-15 %. Ozonesonde and COH biases are similar for the two periods. Aura MLS bias is also similar to the COH bias. The M2GMI model comparisons show a larger bias. The GMI CTM shows the smallest bias in comparison to Umkehr

profiles in layers 5-8, whereas the bias increases in the second period but remains the smallest in the upper stratosphere. In layers 2 and 3, GMI CTM has the largest bias, where M2GMI shows the lowest bias. Ozonesondes have the lowest bias in layers 3 and 6, high bias in layers 4 and 5 and large negative bias in layer 2. The models have lower bias in layers 6-8 as compared to observations (satellite and ozonesonde), and larger bias in layers 2-4. The mean offset is calculated by averaging results from all datasets (4 datasets before 2005 and 5 after 2005), and horizontal bars represent the standard deviation of the

mean values.

Panel b of Fig. 1 shows comparison of the same datasets, but Umkehr profiles are processed using standardized stray light corrections (SLC, Petropavlovskikh et al, 2011). It is found that SLC reduces bias in layers 7, 8 and 9, increases biases in layers 4 and 3 (GMI bias in layer 3 becomes the largest among all layers), whereas biases in layers 5, 6 and 1 do not change significantly.

Panel c and d of Fig. 1 summarizes the uncertainty of the bias calculated for operational and SLC Umkehr profiles respectively. The solid (dashed) lines show results for 2005 - 2018 (1995 -2004) comparison periods. There is no large difference found





between standard deviations (SD) in two time periods, and they are larger than 5 % in layers 2, 3 and 4. The largest SDs are found in comparisons between ozonesonde and Umkehr. This could be related to a large vertical variability captured by ozonesondes and the limitations in the Umkehr AK smoothing. However, the SD in layer 2 is still below 15 %, which is the

estimated Umkehr retrieval uncertainty in the bottom layers. In summary, we demonstrated that the standardized stray light corrections do not fully reduce the bias between Umkehr and other ozone observing methods. Since the optical characterization of each Dobson instrument is not yet possible, the optimization approach is discussed next. In this paper we discuss an empirical approach to minimize simulated and observed Umkehr differences at large SZAs.

### 3.3 Empirical correction methodology

This section describes the new method developed for optimization of Dobson ozone profile retrievals to account for the instrument-specific out-of-band stray light and other optical artifacts. This approach is used for homogenization of the long-term Umkehr records. The corrections for each instrumental record in the station time series are developed to remove artificial steps in the NOAA Umkehr ozone profile records and to reduce the bias relative to other ozone observing systems. To minimize instrumental artifacts in Umkehr observations (unknown instrumental optical degradation or contribution of the background

noise) the Umkehr retrieval forward model (simulation of the observation) is forced to match the auxiliary or reference ozone profile. For example, the M2GMI ozone and temperature profiles simulated near the location of Dobson station in Boulder are assumed to represent atmospheric absorption and molecular scattering properties (assuming no aerosols in the atmosphere) for the day (and time) of the Umkehr measurement. The use of daily changing temperature profiles modifies the ozone absorption cross section that allows an improvement in the model fit to the day-to-day variability in N-values at large SZAs (see Appendix

C for further discussion). The forward model of the Umkehr retrieval first calculates the single scattering zenith sky intensities with high (0.1 nm) spectral resolution. The convolution of spectrally resolved zenith sky radiances and standardized band-pass functions (Komhyr, 1993) are performed to create N-values at ten nominal SZAs. In the next step, the multiple scattering and refraction corrections are selected from look-up tables (LUT) that are prepared by the radiative transfer simulations of the Umkehr observations (Petropavlovskikh et al, 2005; Petropavlovskikh et al, 2009) using a set of climatological ozone profiles

(McPeters et al, 1998). Corrections are selected based on the station location (i.e. in low, middle or high latitude regions) and adjusted to the total ozone observed for the day. In the following step the standardized stray light out-of-band corrections are selected from LUT similarly developed to the scheme described above (Petropavlovskikh et al, 2011). This means that up to this point the Umkehr N-values are simulated for an idealized Dobson instrument. The assumption for out-of-band rejection (or SLC) of the UV light in a typical Dobson instrument is on the order of 2 x10$^{-5}$ level (Petropavlovskikh et al., 2011) but can

vary between instruments (Moeni et al, 2019) and therefore can vary between Dobson instruments sequentially operated to create the long-term station record.

In order to test the representativeness of the M2GMI's vertical ozone distribution over Boulder, the above described process is repeated by using several reference ozone records, including Boulder overpass output from the GMI CTM and M2GMI models, FG11ap and QBOap climatology, and combined MLS and ozonesonde profiles (SND_MLS) matched to Umkehr





station location and date of observation (see data description in Section 2). Differences between simulated and measured Umkehr N-values are averaged over the time between two consecutive calibrations of the Dobson instrument at each nominal SZA to create an empirical correction for the Umkehr curve simulated by the forward model. This unique correction is applied to re-process each Umkehr measurement (AM and PM separately) taken during the reanalysed time period, and the new ozone profile is called optimized. Note that optimized ozone retrieval includes both the standardized SLC and the new empirical

instrumental correction. The homogenized time series is created after all individual observational periods are reprocessed (see Section 4.2 for further discussion).

Table 2 contains the dates and time periods selected to apply empirically derived adjustments to Dobson observations in Boulder, CO. The decision to adjust Umkehr data is tested every time the station instrument is replaced with a new instrument.

The change in observations can occur due to different levels of out-of-band rejection unique to each Dobson instrument optics system. Therefore, another reason to re-process the data is after optical repair whether caused by sudden physical damage (i.e. fall of the instrument) or long-term wear-and-tear due to exposure to the weather elements (i.e. sea salt erosion). The instrument repair can include replacement of the optical wedge, replacement of the photo-counter or change of the centre of the bandpass due to a new temperature setup for the Q-levers).

The optimization method accounts for undetermined deviations in the optical system that have not been captured at the time of the exchange or repair of the Dobson instruments. The changes may not be significant for accuracy of the total column ozone observations, but may be large enough to change Umkehr curve and create a step change in ozone record. To verify empirical adjustments and the consistency of re-processed Umkehr time series, in section 5 we present comparisons with independent ozone observing systems (satellites, ozonesonde) and co-incident with Dobson observations.

**3.4 Discussion of optimization results**

**Figure 2a** summarizes adjustments to the simulated N values that are needed to match Umkehr observations in Boulder with other reference records between 2005 and 2018. Examples of several N-value corrections are shown for Umkehr simulations where ozone profiles from several datasets were used as the reference ozone profile information. The daily differences are averaged over 2005-2018 period and plotted at each Umkehr nominal SZA . The mean N-value correction and standard

deviations are shown as grey boxes. The measurement uncertainty of typical Umkehr observations range between 0.5 N-value at 70-degrees SZA and up to 1.2 N-value at 90-degrees SZA (i.e. standard deviations of the error covariance matrix). The empirical corrections appear to agree within the observation uncertainty. However, the largest negative correction at 86.5 degrees SZA varies between -0.1 and -1.4 N-values depending on the reference dataset. Also all corrections exhibit similar shape with respect to the SZA. Figure 2b summarizes distribution of Umkehr N-value residuals calculated with M2GMI

reference profiles. The width of the distributions increases with SZA. The largest deviations are found when the QBOap dataset is used as a reference (not shown) and the lowest deviations are found when the M2GMI ozone profiles are used as a reference.



In order to select the most effective empirical adjustment for the Boulder Umkehr data processing in 2008-2015, Umkehr ozone profiles retrieved with multiple empirical corrections are compared to the MLS station-overpass ozone profiles (Fig.2c). The goal is to have a zero bias through the difference profile comparisons with MLS. Empirical optimizations minimize ozone
bias in comparisons to the MLS profiles; however, optimized Umkehr profiles still show +/- 5 % bias, even when the MLS profiles are used as a reference (see results for the MLS and sonde combined profile, SND_MLS). There are some differences between optimized datasets, but they all agree within the uncertainty of each empirical correction. Results show the wave-like distribution of biases that change from negative bias in the upper stratosphere to positive in the middle, then again to the negative bias in the lower stratosphere and to the positive bias in the troposphere. Some of these biases are due to the Rodgers
optimal estimation technique that relies on the vertical ozone profile smoothing and a priori covariance that assumes cross correlations between adjoining layers (Rodgers, 1990). There is also a limitation in Umkehr observations that makes it difficult to clearly separate ozone information between tropospheric and stratospheric layers.

Since no Aura MLS data are available prior to 2004, we select the M2GMI dataset to develop optimized corrections for the entire Umkehr record. The M2GMI correction is derived separately for each calibration time period of the Dobson record in
Boulder (Table 1). We note that the M2GMI-based optimized correction produces a small (+/- 5 %) but significant bias in retrieved Umkehr ozone profiles relative to the MLS profiles averaged over 2005-2018 period (see Figure 2c, green line). It means that there is an additional difference between the atmospheric state and the Umkehr observation that is not adequately simulated in the forward model of the Umkehr retrieval. Therefore, the iterative modification of the M2GMI N-value correction (see dark line in Fig.2 a) is performed until the remaining bias between optimized Umkehr (M2GMI) and MLS is minimized
in the 2005-2018 period. The final adjustment for 2005-2018 period is marked as M2GMI* (dark thick curves in panel a and c). This additional correction to M2GMI optimization is applied to all M2GMI empirical corrections prior to re-processing the entire Umkehr record.

Figure 3 summarises the time-series of all empirical corrections as a function of SZAs applied to the Umkehr observations in Boulder after 1994. Three panels show the SLC (a), M2GMI empirical corrections (b) and the combined correction (c). The
black arrows at the bottom indicate dates of Dobson calibrations and/or instrument replacements (see Table 2 for the dates). The optimized corrections indicate several distinct time periods that change the mean ozone levels in time series and therefore impact trends calculated after 2000.

## 4. Comparisons of optimized Umkehr time series against reference records

This section discusses the vertical and temporal changes in the Umkehr optimized ozone record. Comparisons between
operational OPR (red), standardized SLC (green) and optimized OPT (blue) ozone in layer 8 (4-2 hPa) at Boulder are plotted in Fig. 4 as a function of time. The biases between Operational and SLC (magenta) and SLC and Optimized (black) demonstrate the main temporal difference between the time series. It is apparent that the SLC time series features an additional seasonal cycle that is total ozone dependent and corrects the out-of-band straylight errors in operational Umkehr record. The optimized


Umkehr version in addition to the SLC uses M2GMI-based empirical corrections developed for each period marked by blue arrows.

The offsets between OPR vs SLC versions vary from -5 % (in the period since last calibration) to up to -10 % in 1994-1998 and 2005-2009 periods, and -7 % bias in 1999-2004 and 2009-2012 periods. The up and down biases in the optimized record impact the linear trends in layer 8 ozone (reduced by ~ 2 % per decade based on a simple linear fit) as compared to those derived from the operational or SLC version of Umkehr record. The replacement of Dobson 82 with Dobson 61 resulted in

step change in Umkehr ozone in layer 8 at the beginning of time series (see Figure B3 showing 1979-1994 period). The optimization method identified the need for an adjustment on the order of 10 %. This change in stratospheric ozone levels at the beginning of the Boulder Umkehr record can significantly reduce trends derived from the homogenised record (optimized series) prior to 1997 and bring it to a closer agreement with the satellite combined zonally averaged trends (LOTUS report, Figure 5.9a in Chapter 5). A discussion of trends is beyond of the focus of this paper and will be addressed in a follow up

publication.

The step changes in the differences are clearly seen in this plot and vary between 0 and -15 % during non-volcanic periods, while during the volcanic periods (see Fig. B3) corrections can be as large as -30 %. Optimizing Umkehr ozone profile retrievals during the volcanic eruption follows a similar procedure as described above. When large volcanic eruptions inject aerosols into the stratosphere the operational Umkehr retrieval is not set up to account for the change in atmospheric scattering.

Therefore, the errors in operationally retrieved Umkehr profiles can be as large as 70 %. For trend analyses, the volcanic time periods in the Umkehr time series (i.e. 1991-1993) are typically removed prior to fitting the statistical model to the data. The optimization method reduces the introduction of gaps in Umkehr time series so that the entire record can be used for trend analysis. An example of volcanic period corrections and discussion of results is shown in Appendix B.

### 4.1 Changes in mean and seasonal biases

After reprocessing of the Umkehr data with optimization corrections, the changes to vertical profiles are verified through comparisons against independent ozone observations that are matched with Umkehr record. For verification of changes in Umkehr data at Boulder, satellite overpass data are used for comparisons (Table 3) and co-incident ozonesonde profiles.

Figure 5 shows Boulder Umkehr data comparisons for the two time periods: 1994- 2004 and 2005-2020. The reference data are the same as in Figure 1.

In comparison to the results shown in Figure 1 (Umkehr operational and SLC versions) the optimization process significantly reduces satellite/Umkehr biases in the upper stratosphere (Umkehr layers 7-9). For example, a small bias (< 2 %) is found between Umkehr and AGG, COH and MLS profiles above 30 hPa. The comparisons with GMI CTM and M2GMI models show 3-5% negative bias that is slightly larger in 1994-2004 period as compared to the 2005-2020 period. In comparison to the operational Umkehr data, the bias is smaller and has changed sign. The agreement between the modelled ozone and the

Umkehr is better for the SLC version rather than the optimized retrieval. However, the optimization is not meant to reduce the



bias between the model and Umkehr. The models are used only as a reference to assure the continuity of optimized ozone after the Dobson calibration.

In the middle/lower stratosphere (layer 3-4, ~125-30 hPa) small positive biases (<5%) are found between satellite, COH, AGG and MLS, and optimized Umkehr records. Also positive biases (up to 8 %) are found in comparisons with M2GMI profiles in

2005-2020 period, whereas the bias is negligible in GMI CTM comparisons. In the UTLS (layer 2, 250-125 hPa) positive biases (5-10 %) are found in comparisons against COH (1994-2020) and MLS (2005-2020) satellite data. However, these biases are within the uncertainty of the Umkehr retrieval in layer 2 (~10 %). Comparisons of optimized Umkehr profiles with the date/time matched ozonesonde data in Boulder show that prior to 2004, the bias from optimized Umkehr is ~ 5% in layer 3-5, but after 2005 the bias is largely reduced.

For the validation of Umkehr optimization it is important to track changes in the reference records (especially models) before and after 1998, and around 2005 as the discontinuity can lead to an introduction of the artificial trend in homogenized Umkehr time series. The step change in GMI CTM ozone record in 2005 was discussed in several recently published papers (i.e. Orbe et al 2017, Stauffer et al, 2019). The bias is due to assimilation of the Aura MLS temperature profiles in MERRA2 starting at the end of 2004. The change in temperature fields affects the winds generated by MERRA2 that drives transport in both

M2GMI and GMI CTM ozone simulations as well as stratospheric chemistry. The biases relative to Umkehr above 30 hPa are almost identical in both models, while small reduction in biases (<2 %) is found after 2005. This is the expected result as ozone variability in the upper stratosphere is largely determined by the chemistry that is similarly described in both models. In the lower stratosphere the transport of the chemicals is driven by the MERRA-2 meteorological fields; however, the modelled ozone profiles are not forced to reproduce the MLS profiles. Note, that Umkehr and MLS satellite overpass comparisons for

2005-2020 period show a smaller bias below 30 hPa than in GMI CTM comparisons, but it is larger than the bias between ozonesonde and Umkehr. M2GMI does not show a significant bias below 30 hPa and compares well against Boulder ozonesonde in 2005-2020 period. The differences between two models are likely related to the differences in ozone mixing across the tropopause as discussed in Stauffer et al 2019 paper. There are small changes in biases between the models after 2005, however COH and M2GMI relative biases to Umkehr show little change after 2005.

Figure 6a shows seasonal differences between the COH overpass record and the operational (left) or optimised (right) Umkehr ozone profiles collected in Boulder. The plots show the monthly percent differences averaged from 2000 to 2018. The optimized version of the Umkehr data shows a significant reduction in biases as compared to the operational version in the layers above 10 hPa, while small positive and somewhat seasonally varying bias (up to 5 %) remains between 100 and 10 hPa. The largest negative bias is found near 200 hPa in winter months. It could be related to the 5-10 % differences in the a priori

data used for satellite and Umkehr retrievals (panel b), and limitations in both systems to the sensing of ozone variability in the lowest stratosphere. To summarize, optimization of the Umkehr observational record in Boulder over 2000-2018 period reduced mean biases between Umkehr and COH ozone in the upper and middle stratosphere. Since SBUV/2 and OMPS ozone information in lower layers are strongly influenced by a priori, and have little independent information there (Kramarova et al. 2013), this result is expected.





## 4.2 Temporal changes in optimized Umkehr time series

**Figure 7** shows 1994-2020 comparisons between monthly mean ozone retrieved by the operational Umkehr processing (black line) and ozone derived after optimization (blue line). Three panels (top to bottom) show comparisons in layer 8 (4-2 hPa), 6 (16-8 hPa) and 4 (64-32 hPa) for Boulder Umkehr record. Similar plots for OHP, MLO and Lauder records are included in the Appendix A. The arrow symbols at the bottom of the plot indicate the dates of the Dobson instrument calibrations or instrument replacements (see Table 2 for the dates of calibrations and the WinDobson automation events). The standardized stray light correction is a long-term mean adjustment that depends on ozone climatology and is total ozone dependent (Petropavlovskikh et al, 2011). It creates the seasonally dependent adjustment (less than 1 % in the upper stratosphere), but this correction does not add significant long-term trend. However, different optimized corrections are applied to the individual periods between instrument calibrations, which results in different amounts of increases in the retrieved ozone.

To highlight changes in optimized Umkehr, the COH overpass ozone record is plotted as a reference (red line). The vertical dotted lines indicate the periods of satellite records (also see abbreviations at the top of the plot) that are combined in the COH ozone dataset. The difference between COH and operational Umkehr data is shown as a dark green line with the mean negative bias of ~15 % (0%, 5 %) in layer 8 (6, 4 respectively) that varies seasonally and temporally. The percent difference between optimized Umkehr and COH is shown as a light green line. The average bias is close to zero, while the seasonal changes are also reduced in layer 8 and 4. The main change in the optimized Umkehr ozone dataset is the increase/decrease in ozone amount vertically (three panels in Fig. 7). Also noticeable are changes in the relative shifts between calibration periods. For example, the change in the offset between COH and operational Umkehr biases (dark green line) is seen in 2001-2006 and 2006-2011 periods. This step change offset is largely reduced in COH comparisons against the optimized Umkehr version (light green). In addition, we do not find any evidence of a step change in the optimized data in 1998 or 2004/2005 that could be related to step-changes found in M2GMI (Stauffer et al, 2019). Similarly, in optimized records of three other Umkehr stations (see Appendix A) we do not find any impact from the M2GMI step changes. For example, at MLO station, the instrumental artifacts in Dobson operations resulted in significant step change in Umkehr operational data but were completely eliminated by the optimization method (see Fig. A6a.). The importance of these changes for trend analyses is alluded in the LOTUS 2019 Report, and will be further discussed in a future paper.

## 5. Conclusions

In this paper we discussed a method for the Umkehr profile optimization and its impact on the homogenization of the long-term records. The lack of the optical characterization of Dobson instruments used for Umkehr method observations results in the biases in the retrieved ozone profiles relative to other observing systems. The previous approach of standardised stray light corrections helped with reduction of biases but did not eliminate the step changes in the station record associated with instrumental changes. The optimization method relies on the experience of Dobson operators, knowledge of the Dobson World calibration centre (operated by NOAA GML in Boulder for more than 40 years) and its records of instrument calibrations. The





careful and robust approach to instrument exchanges, repairs and calibrations against the WMO standard Dobson 083 allowed for collection of high quality long-term records of stratospheric ozone changes. The optimization provides a tool for a fine-tuning of the Umkehr retrievals, removing of the instrumental biases, and empirically evaluating the impacts of stray light

contributions to the observations over different time periods. However, the optimization is not meant to reduce the bias between the reference model and Umkehr ozone profile. The models are used only as a guide to assure the continuity of optimized ozone after evaluating and removing step changes caused by Dobson instrumental artifacts, changes to the data collection protocols and data processing. This careful approach aims at homogenizing Umkehr time series for trend analyses, reducing noise in the data and supporting NOAA and WMO efforts at detection of ozone recovery under the Montreal Protocol guidance.

**Appendix A. Umkehr biases and time series for OHP, MLO and Lauder**

This appendix shows comparisons of the operational (Fig. A1), SLC (Fig. A2) and optimized (Fig. A3) versions of Umkehr records from OHP, MLO and Lauder against multiple independent records during the 2005-2019. The biases between operationally processed Umkehr data and other records (including ozonesonde station records) are similar in vertical distribution (i.e. large positive biases in the upper and middle stratosphere and negative biases in the UTLS) to the results of

Boulder Umkehr record. A large noticeable bias between M2GMI and GMI CTM ozone profiles is found at Lauder station in the lower stratosphere (layers 4, 3 and 2). The SLC versions of the Umkehr data at three stations (Fig. A2) show reduction in biases in the upper stratosphere similar to the changes found in Boulder record (Fig. 5a). This reduction in upper layer biases is accompanied by changes in the biases in the lower layers, especially in layer 2 (decrease), 3 (increase for OHP and Lauder) and layer 4 (increase for MLO).

After the optimization (Fig. A3), a significant reduction in biases is found in layers 4-9 with the exception of comparisons at Lauder where M2GMI bias has become more negative, but still less than 5 %, which is within the uncertainty of the Umkehr retrieval. At the same time, the M2GMI bias in layer 2 increased to 5 % at OHP and less than 5 % at MLO, which is similar to the bias between ozonesonde and optimized Umkehr. The Lauder station features a negative bias between the optimized Umkehr and ozonesonde data in layers 2 and 3, while a positive bias of similar magnitude is found at OHP. No significant

biases are found in comparisons of MLO optimized Umkehr with Hilo ozonesonde data. The Hilo ozonesonde profiles are limited to the pressure level above 680 hPa, which is the surface pressure at MLO.

The Umkehr and COH comparisons in layer 2 and 3 at all three stations show small and positive biases (<7 %). The AGG biases with respect to optimized Umkehr are similar to the COH-Umkehr biases, which suggests that AGG and COH records compare well during 2005 to 2019 periods. The largest bias is found between GMI CTM and optimized Umkehr at Lauder and

OHP locations. The MLS record agrees with other records at OHP and MLO, but is biased high in layer 4 and agrees with COH and AGG data in layer 3, while in layer 2 MLS bias is similar to ozonesonde and M2GMI records.

The impact of standardized SLC and optimized corrections derived for OHP (Fig. A4), MLO (Fig. A5) and Lauder (Fig. A6) are comparable to the results shown in Figure 3, where the largest changes to N-values are found at larger SZAs. The





optimization (panel c in Fig. A4 and A5) are coincident with Dobson calibration (signified by black arrows at the bottom of
the panel), that is specific to the history of the instrument operations at each station.

In order to show time series of changes in Umkehr record after optimization at OHP, MLO and Lauder stations, we included
Figures A7, A8 and A9 that represent similar content shown in Figure 7. These Figures show monthly average ozone time
series derived from the operational and optimized Umkehr versions, and the COH satellite coincident record. The bottom panel
show comparisons between Umkehr and COH. The arrows at the bottom of the panel point to the dates of Dobson calibrations
and Umkehr changes after optimization process specific to each station. Results in these Figures are presented in three panels
corresponding to ozone variability layers 8, 6, and 4 respectively. The largest changes are found in Layer 8, whereas ozone in
layer 6 and layer 4 shows the smallest impact of the optimization process as expected from the vertical contribution of the out-
of-band errors.

The temperature related step-change in the upper stratosphere in the MERRA-2 assimilations is reflected in both M2GMI and
GMI CTM models (Stauffer et al, 2019 and references therein). The step change in MERRA-2 is from the addition of MLS
temperature to the observing system starting in 2004. Ozone chemical timescales are short in the upper stratosphere, so any
transport induced changes have little to no impact on ozone. However, temperature impact kinetic reactions rates. Therefore,
the step change in assimilated temperatures creates a step-change in stratospheric ozone.

To assess the impact of the 2004/2005 step-change in the M2GMI and GMI CTM records (Stauffer et al, 2019) on Umkehr
optimization and homogenization process we plot biases for 1994-2004 and 2005-2020 periods (Fig. A10). Based on these
comparisons, we notice that the biases between M2GMI and Umkehr are reduced by 2 % in the upper stratosphere in 2005-
2020 period, whereas relative biases between M2GMI and GMI CTM are contained. Similar changes in the upper stratospheric
biases between the M2GMI and optimized Umkehr for two analysed periods are found for Boulder (Fig. 5a), although the
absolute biases are different. The 2005-2020 period shows an improved agreement between Umkehr and M2GMI record, as
640 well as bias between Umkehr and COH record is reduced below 20 hPa, except there is no reduction with respect to the GMI
CTM bias. The differences in stratospheric ozone offsets between ozonesonde/M2GMI (smaller) and ozonesonde/GMI (larger)
over Lauder were reported in Stauffer et al. (2019) paper and had similar range of biases derived in this study at the lower and
middle stratosphere.

In conclusion, the optimization reduces biases between Umkehr ozone profiles and most of the alternative coincident records
645 to less than +/- 5 % at all four stations. Investigation of large biases between GMI CTM and other records at Lauder needs
further investigation.

**Appendix B. Umkehr optimization for volcanic time periods.**

During the volcanic eruptions (like El Chichon in 1982 or Pinatubo in 1991) sulphate aerosols were ejected to stratosphere and
were transported globally or semi-globally. The large amounts of aerosol load, as large as to 0.1 in optical depth in UV
650 wavelengths (Stevermer et al, 2000), significantly contributed to the scattered light in the atmosphere. Figure B1, panel a
shows two large deviations during the volcanic period in the apparent transmission time series measured at MLO. While the



operation Umkehr ozone retrieval algorithm does not account for aerosol-produced scatter in its forward model, and therefore interprets changes in observed N-values as changes in ozone profile. This creates a period of erroneous ozone values. Figure B1, panel a, shows the reduced ozone in stratospheric layer 8 of the MLO operational Umkehr record (black line). The most depleted ozone is found soon after the eruption, coincident with the largest aerosol load, and then the error in retrieved ozone is slowly reduced following a decay in aerosol particles over ~2-year time period.

The optimization to remove the low bias in the Umkehr ozone profile during volcanic periods is performed following a similar approach as discussed in this paper, except the corrections are developed for several 6-month long incremental periods. The result is shown in Panel b with a blue line. For the reference, the M2GMI data are also shown (red line) and compare well with the optimized Umkehr results. Yet as an independent reference the SAGE II data are also shown for comparisons (green line). Volcanic corrections to the Umkehr N-values are presented in panel c of Figure B1 for three stations (Boulder, MLO and Lauder) as function of SZA and time. The timing of the volcanic aerosol transport and decay can be discerned at tropical latitude locations as captured in the Umkehr record at MLO in 1991-1993. While in the northern middle latitudes (Boulder) the aerosols appear to be still impacting Umkehr observations at large SZAs until 1997, and in the southern hemisphere (Lauder) until 1999.

[Describe the rest of the figures? Note if you compare to COH, you should note that COH as also affected by volcanic aerosols. Should not be used at least at lower levels, questionable at higher levels. Comparisons would be better with SAGE II here.]

**Appendix C. Umkehr Averaging Kernel**

Table C1 summarizes the Umkehr layer system. Each Umkehr layer is defined by the pressure at the bottom of the layer. The highest layer extends to the top of the atmosphere. The standard Umkehr output is constructed in 10-layer system (2 left-most columns in Table C1). The 16-layer system (two middle columns in Table C1) is used for the AK output. The 61-layer system (two right-most columns in Table C1) is the working grid of the Umkehr retrieval algorithm to avoid interpolation errors (Petropavlovskikh et al., 2005).

Figure C1 shows Umkehr Averaging Kernels (AK) as function of pressure and respective Umkehr layers (as defined in Table C1). The 61-, 16- and 10-layer AKs are shown in panels a, b, and c respectively. The AK concept is described by Rodgers (2000). For plotting purposes, we follow Bhartia et al. (2013) formulation of the "smoothing" kernels that act as the low-pass filters to smooth fractional anomalies in each layer. The Fig. C1 shows the rows of the fractional AK that indicate the sensitivity of the retrieved ozone at that layer to changes in ozone at all layers. The red/green colors (see legend in panel a) highlights the high/low informational content of the AK. The maximum AK values between ~100 and 2 hPa are aligned with the nominal altitude of the layer, indicating the highest informational content obtained from that layer, while contribution from adjacent layers is reduced at an exponential rate. Below 100 hPa (and above 2 hPa) level the maximum of the AK is shifted higher (lower) in altitude and the AK becomes broader. This means that the retrieval is the most sensitive to the ozone variability in the above (below) layers and therefore relies more heavily on the a priori information in order to separate



informational content into individual layers. The Umkehr ozone profile is reported in 10 layers selected such that the
informational content is provided at about two datapoints per width of the smoothing kernel. The vertical resolution at the
bottom of the profile is poor and therefore several layers are combined into a thicker layer that represents tropospheric column
ozone information (see Table C1).

**Appendix D. Temperature sensitivity in Umkehr retrievals**

The UMK08 operational algorithm (Petropavlovskikh et al. 2005) is based on the Bass and Paur (BP) ozone cross-section
(Bass and Paur, 1985), convolved over the Dobson C-pair standardized band-pass (Komhyr et al, 1993; Petropavlovskikh
et. al, 2011). The impact of the ozone cross-section on the uncertainty of Brewer- and Dobson-observed total column ozone
retrieval was outlined in detail by Redondas et al. (2014). They found that the use of Serdyuchenko et al. (2013) cross-section
reduces the bias between Dobson and Brewer total column ozone observations and also reduces the temperature-dependent
biases. An ad hoc commission of the Scientific Advisory Group (SAG) of the Global Atmosphere Watch (GAW) of the World
Meteorological Organization (WMO) and the International Ozone Commission IO3C) of the International Association of
Meteorology and Atmospheric Sciences (IAMAS) performed the assessment of different ozone cross-section spectral
databases and their temperature dependence (Orphal et al., 2016). For operational Umkehr retrievals, the effects of the ozone
cross-section were found to be minimal (less than 2 %) when NRL (Summer et al 1993) climatological temperatures were used
in the retrieval. However, the report suggests the use of the Serdyuchenko et al. (2013) ozone cross-sections in Umkehr
retrievals whenever total ozone is processed with that cross-section. It is also suggested that day-to-day and diurnal variability
in stratospheric temperatures is larger than represented by climatology and thus may add additional 1-2 % change in daily
stratospheric ozone that is not currently captured by operational Umkehr retrievals.

The nominal absorption cross-sections for the WMO GAW Dobson total ozone network are derived based on the selection of
a standardized temperature −46.3 °C (Komhyr et al, 1993, Redondas et al, 2014). The previously published approach to
represent temperature sensitivity in ozone absorption cross section (i.e. Redondas et al, 2014 and references within) is to use
a second-degree polynomial in temperature. We use a spectrally resolved dataset (Bass and Paur, 1984; Serdyuchenko et al,
2013) to calculate the effective ozone absorption cross-sections and their temperature dependence for C-pair spectral channels.
Next section shows examples of Umkehr retrieved ozone sensitivity to the variability in ozone absorption cross section. **Table
D1** provides coefficients for second-degree polynomials fitted to the Bass and Paur (1984) absorption cross sections over
spectral bands of Dobson Umkehr C-pair (short and long) (Redondas et al., 2014).
**Figure 3** shows a change in Umkehr layer 8 ozone (4-2 hPa) as a function of temperature. The blue dots represent changes
derived from the polynomial fit (Table D1). The dark circle shows the averaged ozone change associated with replacing
standard temperature with the effective temperatures calculated from the subset of the GMI CTM temperature and ozone
profiles matched to Boulder, CO location and selected for 2005-2018 period (GMIave). The red circle represents results based



on the mean temperature from the NRL climatology, 35-46 N (NRLav (N40)), Finally, the dark purple circle shows results based on the ML Climatology at 40 degrees N (McPeter and Labow, 2011). The horizontal whiskers represent a range of seasonal variability in temperature weighted by ozone profiles (between -30 and -12 degrees C) that results in layer 8 ozone changes between -4 and +2 %.

We further assess the representativeness of the NRL temperature climatology used in the operational Umkehr algorithm. We compare the temperature seasonal cycle derived from the M2GMI dataset over the Boulder station to the NRL climatology at 40 degrees N over the 2010-2017 time period (**Figure 4a**). Monthly averaged M2GMI temperatures at 2.8 hPa are shown for the morning (red symbols and lines) and afternoon (grey symbols and lines) Umkehr measurements (i.e. at 15 UTC or 08 LT, and 03 UTC or 20 LT). The NRL temperature climatology is shown for comparison (blue crosses). The day-to-day variability

in M2GMI temperatures are shown as whiskers (the minimum and maximum of all data, one standard deviation above and below the mean of the data). Figure 4a shows that the NRL climatological mean temperature is biased higher in the morning than in the afternoon by ~8.9 and 6.3 K respectively, which is an indication of the diurnal cycle at 2.8 hPa that is not captured by the NRL climatology). Also, day-to-day variability in the stratospheric temperature (see **Figure 4b**, results are based on M2GMI temperature dataset) can create an offset that varies seasonally (i.e. box and whiskers values). The summer months

show less variability in daily temperatures as compared to the winter months, where maximum offset can vary between -5 and 20 degrees C from the NRL climatological.

Impact of daily temperature variability on retrieved Umkehr ozone profiles are further tested for 2015-2017 Boulder Umkehr record. **Figure 5a** shows M2GMI daily and NRL monthly temperatures at 2.8 hPal. The difference in temperatures during days when Umkehr observations were made is also plotted and highlights large deviations in the winter months. This day-to-

day temperature variability as high as 20 degrees results in relatively low ozone variability. The changes in the retrieved ozone are based on temperature sensitivity in the spectrally resolved ozone cross section within Dobson spectral channels and profile smoothing. Panel b of Figure 5 shows comparisons between ozone retrieved using NRL climatology (y-axes) and M2GMI daily temperature profiles (X-axes). Each point is monthly averaged ozone for 2015-2017 time period. The solid line is the 1:1 reference. The mean bias is -0.26 % with 0.3 % standard deviation. This test provides an averaged uncertainty of the daily

retrieved Umkehr ozone in layer 8. It may be a small error; however, this additional uncertainty varies from year to year and thus can have an impact on the long-term ozone trend results in the upper stratosphere.

*Data availability:* The datasets used in this study can be downloaded from https://gml.noaa.gov/aftp/data/ozwv/Dobson/AC4/Umkehr/


*Author contribution*. IP and KM analyzed the data and wrote the manuscript. KM, AM, SS, KW, RQ, GA, and SG prepared the ground-based ozone data, JW, LF, EB prepared satellite ozone datasets for analyses. SS and CW performed the simulations and extracted model data for comparisons with ground-based stations. IP prepared the manuscript with contributions from all co-authors.




*Competing interests.* The authors declare that they have no conflict of interest.

*Financial support*. This study was supported by NOAA Climate Program Office's Atmospheric Chemistry, Carbon Cycle, and Climate program, grant number NA19OAR4310169 (CU)/ NA19OAR4310171 (UMD). IP and JW were also supported

by the NASA SAGE III/ISS award No. NA19OAR4310169. KW is supported by NASA's Global Modeling and Assimilation Office core funding.

*Acknowledgements*. IP and KW would like to thank Luke Oman for developing M2GMI and for very fruitful discussions.

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



**Tables**

**Table 1. NOAA Dobson Umkehr data information: Name of the station, WMO code, dates of the record (month and year), geolocation of the ground-based stations.**

| Location | Site Code | Data Record (MM/YEAR) | Latitude | Longitude | Elevation (m) |
|---|---|---|---|---|---|
| Fairbanks, Alaska | FBK | 03/1984 - 10/2020 | 64.86 N | 147.85 W | 133 |
| Haute Provence, France | OHP | 09/1983 - 12/2020 | 43.93 N | 5.71 E | 685 |
| Boulder, Colorado | BDR | 02/1978 - 12/2020 | 40.02 N | 105.25 W | 1634 |
| Mauna Loa, Hawaii | MLO | 05/1982 - 12/2020 | 19.53 N | 155.58 W | 3400 |
| Perth, Australia | PTH | 03/1969 - 07/2016 | 31.92 S | 115.96 E | 2 |
| Lauder, New Zealand | LDR | 02/1987 - 12/2020 | 45.04 S | 169.68 E | 370 |


Table 2. N-value optimized correction and Dobson calibration history in Boulder. CAL is the period which affected it mainly against C pair in regular calibration of Dobson instrument.

| | | | | | | | Boulder, CO | | | | | |
|---|---|---|---|---|---|---|---|---|---|---|---|---|
| INS | Start Date | SZA 70 | 74 | 77 | 80 | 83 | 85 | 86.5 | 88 | 89 | 90 | |
| 61 | 1993 AUG | 0.0 | 0.2 | 0.5 | 0.6 | 0.4 | 0.4 | 0.9 | 1.0 | 1.3 | 1.8 | |
| 61 | 1999 APR | 0.0 | 0.3 | 0.5 | 0.5 | 0.3 | 0.2 | 0.6 | 0.6 | 0.9 | 1.4 | |
| 61 | 2005 JAN | 0.0 | 0.1 | 0.4 | 0.5 | 0.4 | 0.5 | 0.9 | 1.1 | 1.5 | 2.0 | |
| 61 | 2009 JUN | 0.0 | 0.4 | 0.8 | 1.0 | 0.4 | 0.1 | 0.2 | 0.2 | 0.6 | 1.0 | Updated WD |
| 61 | 2012 SEP | 0.0 | 0.3 | 0.7 | 0.8 | 0.4 | 0.1 | 0.2 | 0.1 | 0.3 | 0.6 | |

| | | | | | | | OHP, France | | | | | |
|---|---|---|---|---|---|---|---|---|---|---|---|---|
| INS | Start Date | SZA 70 | 74 | 77 | 80 | 83 | 85 | 86.5 | 88 | 89 | 90 | |
| 85 | 1994 JAN | 0.0 | 0.4 | 0.5 | 0.6 | 0.7 | 0.9 | 1.0 | 1.0 | 1.2 | 1.6 | |
| 85 | 1996 FEB | 0.0 | 0.5 | 0.8 | 0.7 | -0.1 | -0.3 | -0.2 | -0.1 | 0.4 | 1.1 | |
| 85 | 2000 JAN | 0.0 | 0.5 | 0.7 | 0.8 | 0.6 | 0.5 | 0.4 | 0.2 | 0.3 | 0.8 | |
| 85 | 2007 APR | 0.0 | 0.6 | 0.9 | 1.0 | 0.7 | 0.5 | 0.2 | -0.2 | -0.3 | 0.1 | |
| 85 | 2011 JUL | 0.0 | 0.7 | 1.0 | 1.0 | 0.7 | 0.5 | 0.4 | 0.2 | 0.4 | 0.9 | |
| 85 | 2014 APR | 0.0 | 0.4 | 0.7 | 0.9 | 0.8 | 0.7 | 0.6 | 0.5 | 0.8 | 1.2 | Updated WD |

| | | | | | | | Mauna Loa, Hawaii | | | | | |
|---|---|---|---|---|---|---|---|---|---|---|---|---|
| INS | Start Date | SZA 70 | 74 | 77 | 80 | 83 | 85 | 86.5 | 88 | 89 | 90 | |
| 76 | 1994 JAN | 0.0 | 0.5 | 1.0 | 1.2 | 0.5 | -0.1 | -0.6 | -0.6 | -0.2 | 0.6 | |
| 76 | 1996 JAN | 0.0 | -0.1 | -0.1 | -0.3 | -0.9 | -0.9 | -0.8 | -0.3 | 0.2 | 0.8 | |
| 76 | 2005 JUN | 0.0 | 0.4 | 0.7 | 0.7 | 0.3 | 0.1 | 0.0 | 0.3 | 0.8 | 1.3 | |
| 76 | 2010 JUN | 0.0 | 0.7 | 1.0 | 1.4 | 1.3 | 1.2 | 0.8 | 0.7 | 0.8 | 1.0 | Updated WD |






| INS | Start Date | | SZA 70 | 74 | 77 | 80 | 83 | 85 | 86.5 | 88 | 89 | 90 | |
|---|---|---|---|---|---|---|---|---|---|---|---|---|---|
| | | | | | | Lauder, New Zealand | | | | | | | |
| 72 | 1993 | JAN | 0.0 | 0.0 | 0.0 | 0.1 | 0.4 | 0.7 | 1.0 | 1.2 | 1.3 | 1.7 | |
| 72 | 1996 | JAN | 0.0 | 0.0 | 0.0 | 0.0 | 0.2 | 0.4 | 0.5 | 0.6 | 0.6 | 0.8 | |
| 72 | 1999 | JAN | 0.0 | 0.1 | 0.2 | 0.2 | 0.2 | 0.3 | 0.4 | 0.4 | 0.5 | 0.8 | |
| 72 | 2006 | FEB | 0.0 | 0.0 | 0.1 | 0.2 | 0.1 | 0.2 | 0.4 | 0.6 | 0.8 | 1.0 | |
| 72 | 2012 | FEB | 0.0 | 0.2 | 0.5 | 0.6 | 0.4 | 0.3 | 0.3 | 0.2 | 0.1 | 0.1 | Updated WD |

**Table 3. Satellite ozone profile records used for comparisons with Umkehr records. The time of observational (combined) records and links to the archived records are included for reference.**

| Name | Time Period | References |
|---|---|---|
| COH-NOAA | 1978 - 2020 | ftp://ftp.cpc.ncep.noaa.gov/SBUV_CDR |
| Aggerated NASA (AGG) | 1978 - 2019 | https://acd-ext.gsfc.nasa.gov/anonftp/toms/sbuv/AGGREGATED/ |
| Aura MLS, V4.2 | 2005-2020 | https://avdc.gsfc.nasa.gov/pub/data/satellite/Aura/MLS/V04/L2GPOVP/O3/ |
| SAGEII, V7 | 1984-2000 | https://doi.org/10.5067/ERBS/SAGEII/SOLAR_BINARY_L2-V7.0 |







**Table C1. Umkehr and COH pressure layer grids. The layer is defined by the pressure at the bottom of the layer. The pressure at the top layer is between the pressure level and the top pf the atmosphere. The standard Umkehr output is in 10-layer system (2 leftmost columns). The 16-layer system (two middle columns) is used for the AK output. The 61-layer system (two rightmost columns) is utilized in the forward model.**

| Umkehr 10-layers, Layer # | and Pressure | Umkehr 16-layers, Layer # | and Pressure | Umkehr 61-layers, Layer # | and Pressure | COH 21 layers, layer # | and pressure |
|---|---|---|---|---|---|---|---|
| 1 | 1013.250 | 0 | 1013.250 | 0 | 1013.250 | 1 | 1013.24 |
| 2 | 253.310 | 1 | 506.630 | 1 | 852.038 | 2 | 639.35 |
| 3 | 126.660 | 2 | 253.310 | 2 | 716.476 | 3 | 403.27 |
| 4 | 63.328 | 3 | 126.660 | 3 | 602.482 | 4 | 254.32 |
| 5 | 31.664 | 4 | 63.328 | 4 | 506.625 | 5 | 160.09 |
| 6 | 15.832 | 5 | 31.664 | 5 | 426.019 | 6 | 101.32 |
| 7 | 7.916 | 6 | 15.832 | 6 | 358.238 | 7 | 63.94 |
| 8 | 3.958 | 7 | 7.916 | 7 | 301.241 | 8 | 40.33 |
| 9 | 1.979 | 8 | 3.958 | 8 | 253.313 | 9 | 25.43 |
| 10 | 0.990 | 9 | 1.979 | 9 | 213.010 | 10 | 16.01 |
|  |  | 10 | 0.990 | 10 | 179.119 | 11 | 10.13 |
|  |  | 11 | 0.495 | 11 | 150.621 | 12 | 6.39 |
|  |  | 12 | 0.247 | 12 | 126.656 | 13 | 4.03 |
|  |  | 13 | 0.124 | : | : | 14 | 2.54 |
|  |  | 14 | 0.062 | 40 | 0.990 | 15 | 1.60 |
|  |  | 15 | 0.031 | 41 | 0.832 | 16 | 1.01 |
|  |  |  |  | : | : | : | : |
|  |  |  |  | 59 | 0.037 | 20 | 0.16 |
|  |  |  |  | 60 | 0.031 | 21 | 0.10 |

**Table D1. Wavelengths and temperature coefficients for the second-degree polynomial fit, calculated using B&P ozone cross section data and weighted with C-pair handpasses.**

| C λ | Short | Long |
|---|---|---|
| Band-pass center, nm | 311.5 | 332.4 |
| $C_0$ | 2.196 | 0.1151 |
| $C_1$ | 5.64E-03 | 6.83E-04 |
| $C_2$ | 2.95E-05 | 3.81E-06 |





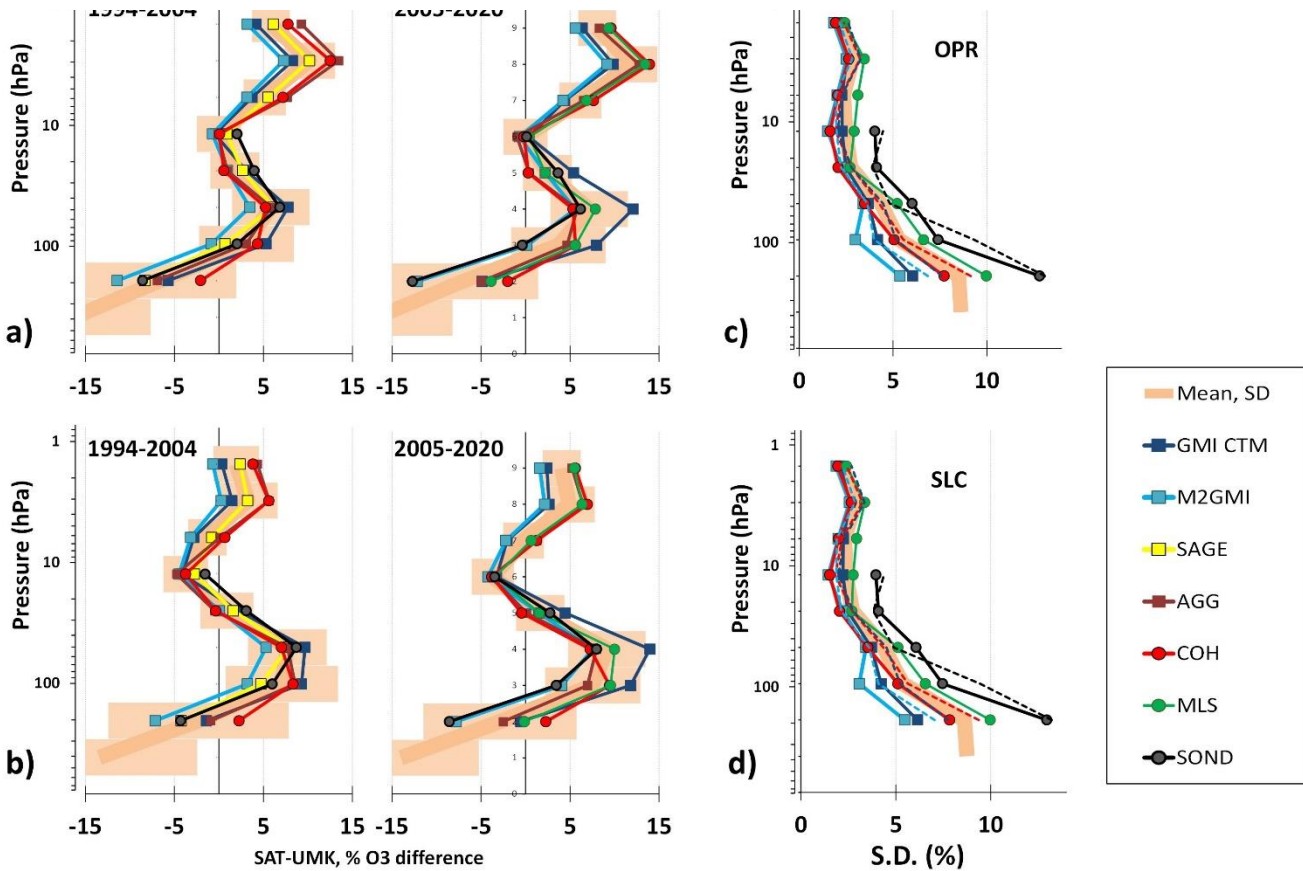


Figure 1. a) Bias between station overpass data from satellite (SBUV/OMPS, Aura MLS), model profile from grid closest to Boulder geolocation (GMI-MERRA, M2GMI), ozonesonde record from Boulder relative to operational Umkehr profiles taken during 1994-2004 (left panel) and 2005-2020 (right panel). Mean profile and SD are calculated as an average (six sets of comparisons) of biases and averaged standard deviations. b) the same as a) but Umkehr retrieval uses standard stray light correction (SLC). c) the same as a) but standard deviation
for mean biases shown in panels a and b. Top) OPR is operational, and Bottom) SLC is standard stray light correction. A solid line is from 2005 to 2020 comparisons, and a dotted line is for 1994-2004 period.



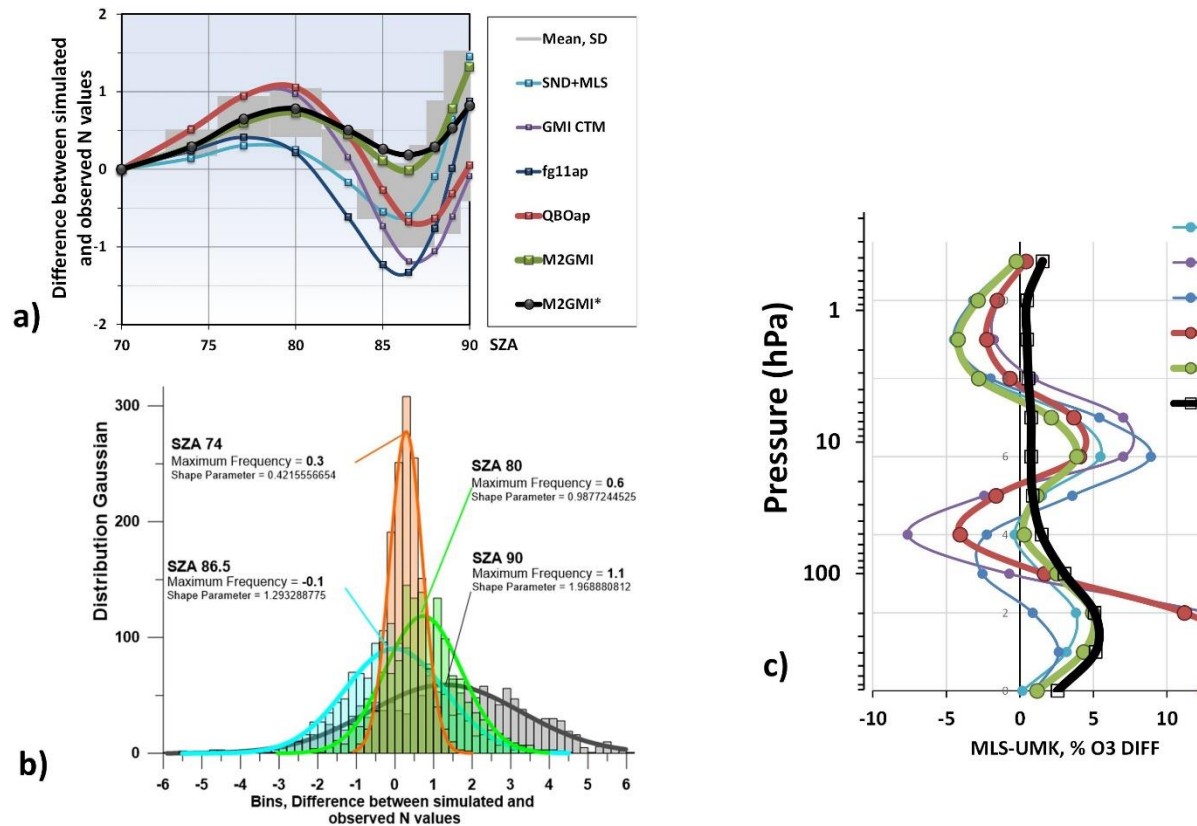

**Figure 2. The simulated ozone profile based on a models, satellites and ozonesonde at Boulder. Difference between simulated N-values and observed N-values. b) the same as a) but M2GMI fits for histograms: Normal Distribution (Gaussian). c) The difference between the MLS and optimized Umkehr ozone profiles with corrections shown in panel a) based on each ozone reference records (see the key for the representation names). The final optimized adjustment M2GMI\* (shown in a) is found through the iterative approach to minimize the bias shown in c).**

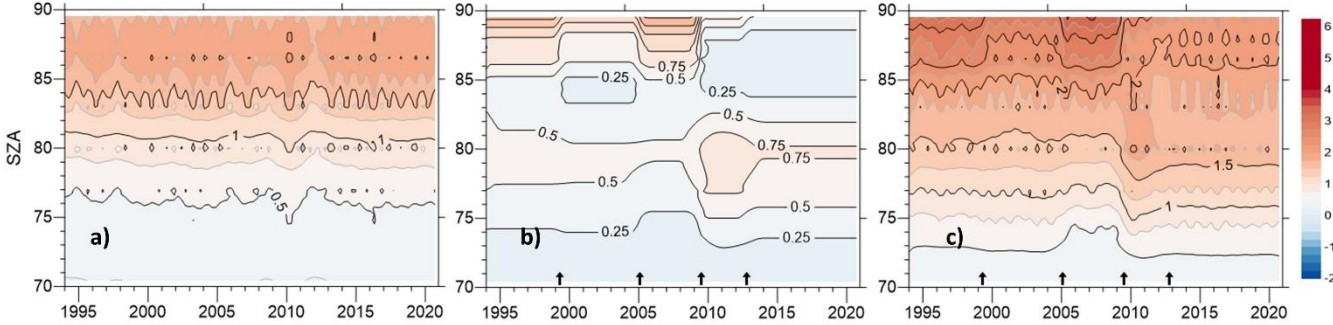

**Figure 3. Plots show the time-series of Umkehr N-value corrections in Boulder: a) Standardized stray light (STL) corrections, b) The optimized stray light correction, c) Final optimization applied by combining STL and optimized corrections. Black arrows at the bottom of panels b) and c) indicate dates of Dobson calibrations and instrument replacements.**





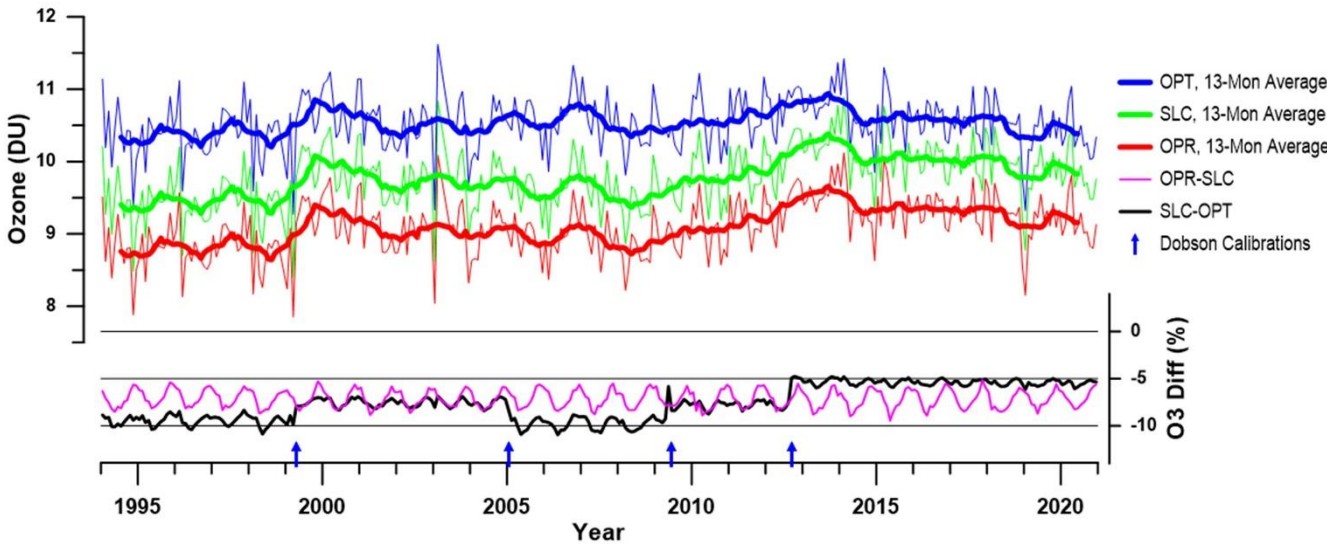

**Figure 4. Upper panel: The time series of Umkehr monthly averaged (thin lines) ozone in layer 8 (4-2 hPa) compare operational (OPR), standard stray light corrected (SLC) and optimized (OPT) versions. The thick lines are 13-months running smoothing. Lower panel: a difference between OPT and SLC (black line), and between OPR and SLC data (purple line).**

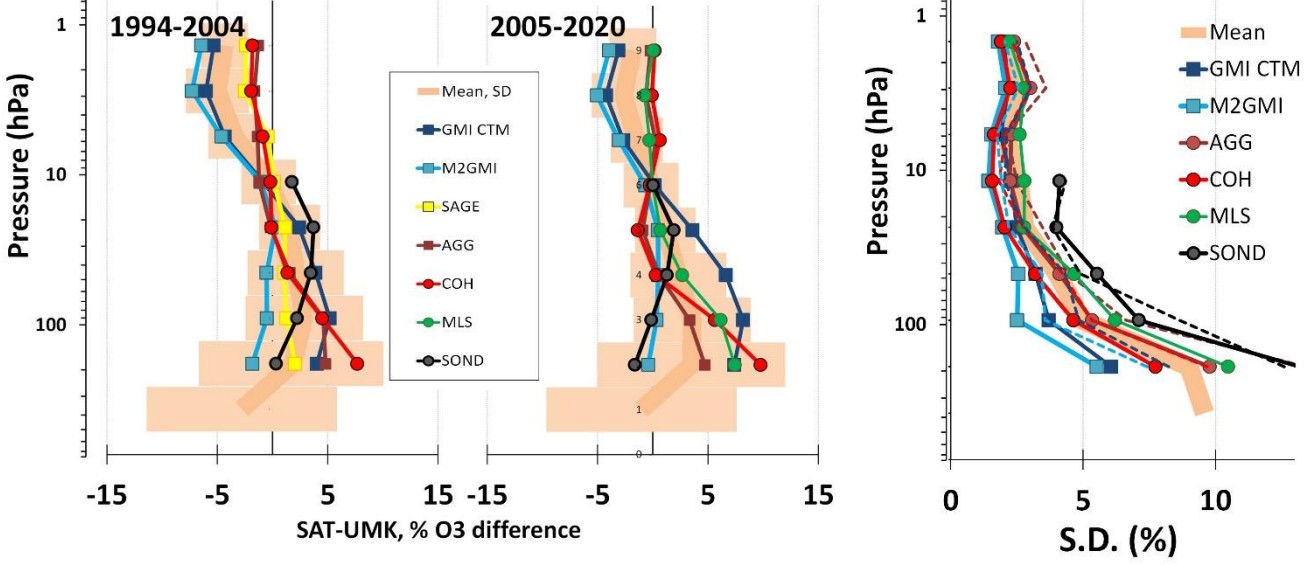

**Figure 5. Same as Figure 1 (old 6) a, but comparisons are against the Umkehr version with optimized stray light correction.**



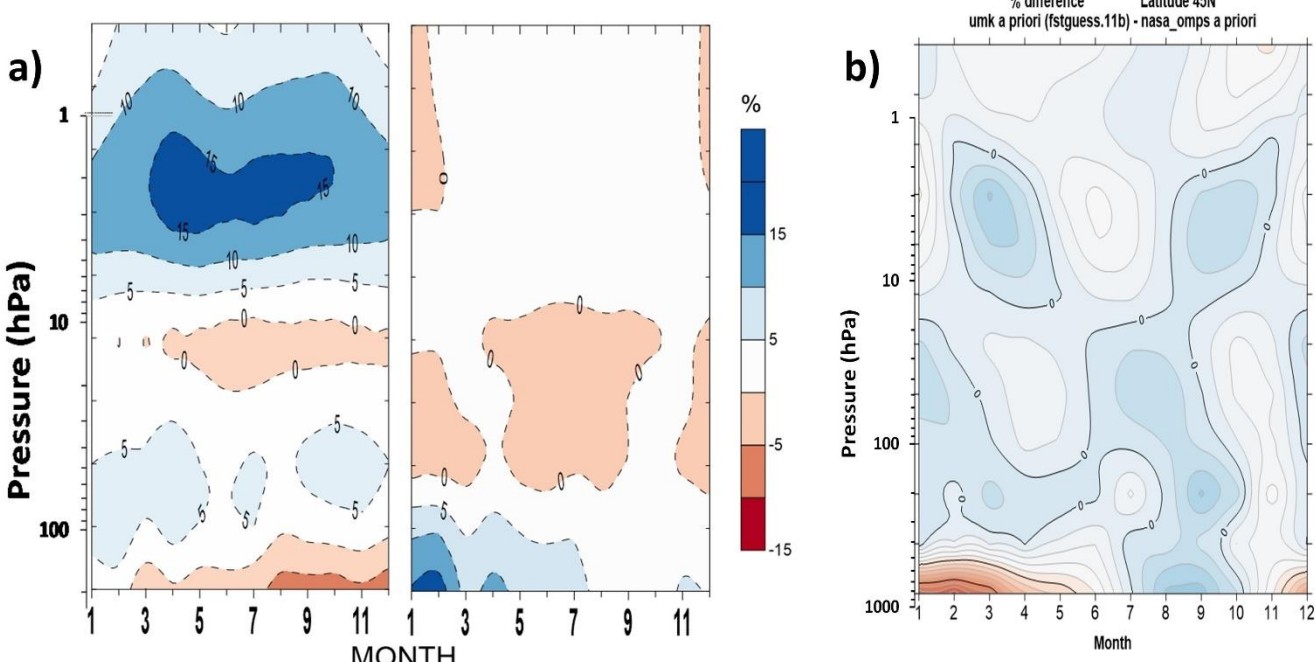

**Figure 6. Seasonal biases between the Umkehr measurements in Boulder and the COH record from 2005 to 2018. Two panels show results for Umkehr retrievals: operational (left), Optimized correction (right). The biases are significantly reduced after the Optimized Umkehr correction. The right panel is the difference between Umkehr 11b (McPeter and Labow, 2011) and OMPS a priori (Flynn et al, 2014).**







Figure 7. The time series of ozone at Boulder in Umkehr layer 8 (2-4 hPa). Operational Umkehr (black), Optimized Umkehr (blue) and COH (red) data are shown as monthly averages. Difference between COH and Operational Umkehr data is shown




as a dark green line. The percent difference between optimized Umkehr and COH ozone is shown as a light green line. The arrows mark  Dobson calibrations and instrument replacement dates. The vertical dotted line indicates begging/end of a series of SBUV/OMPS satellite records that are combined in the long-term COH time series.


Appendix A

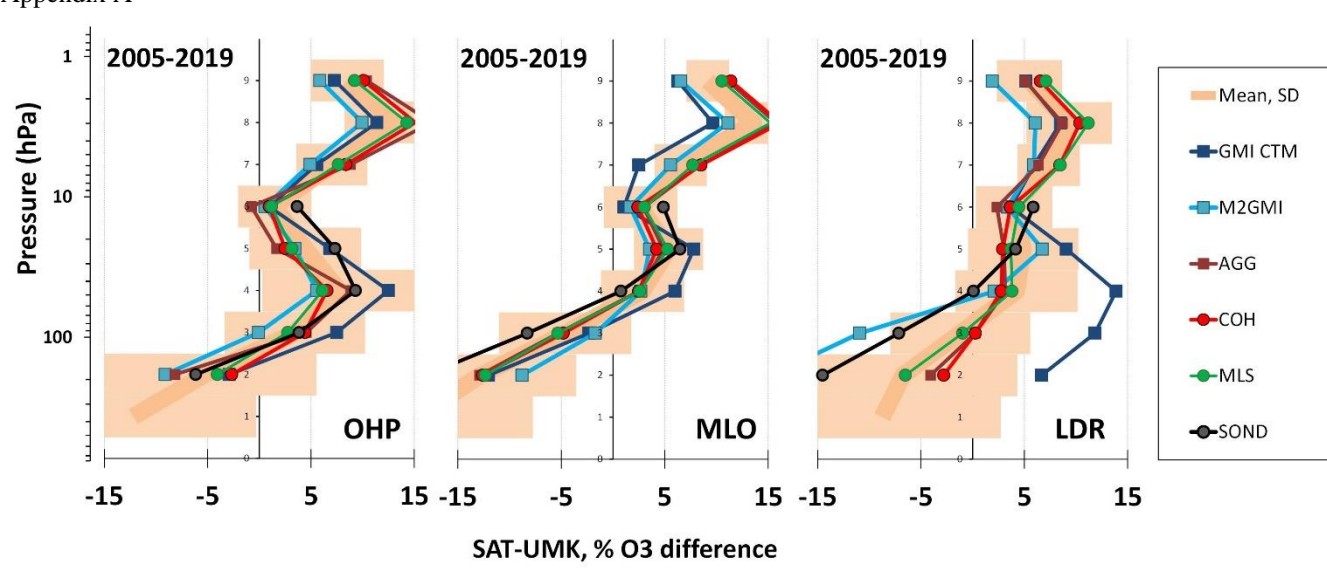

**Figure A1. The same as right panel of Figure 1a, but for three other Umkehr records: OHP (left), MLO (middle) and Lauder (right).**

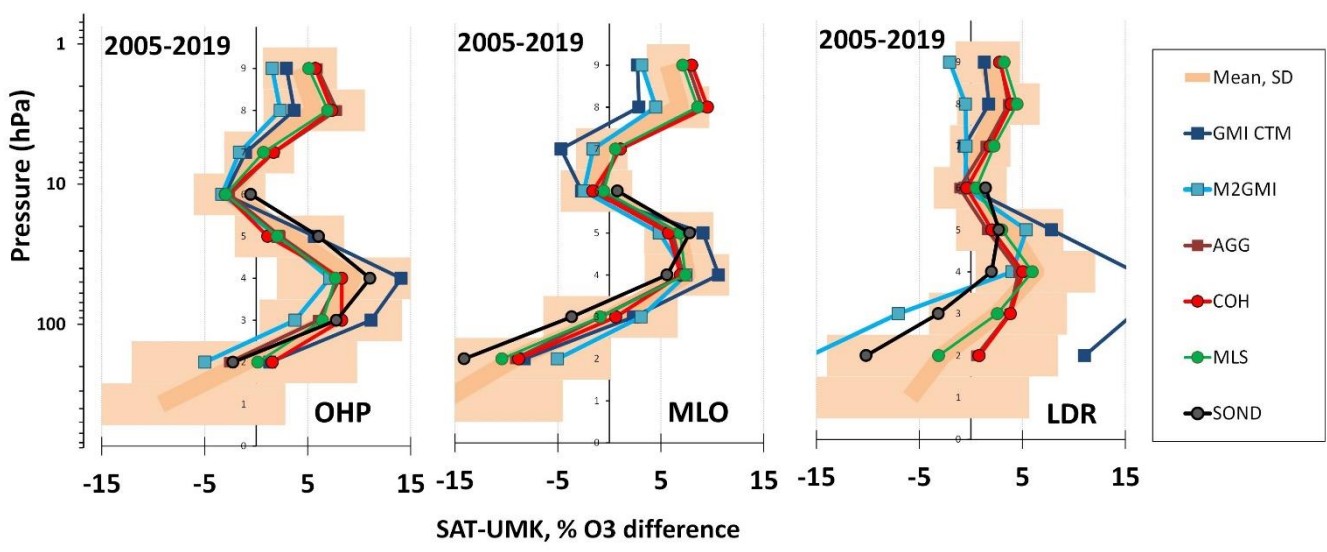


**Figure A2. The same as right panel of Figure 1b, but for three other Umkehr records: OHP (left), MLO (middle) and Lauder (right).**





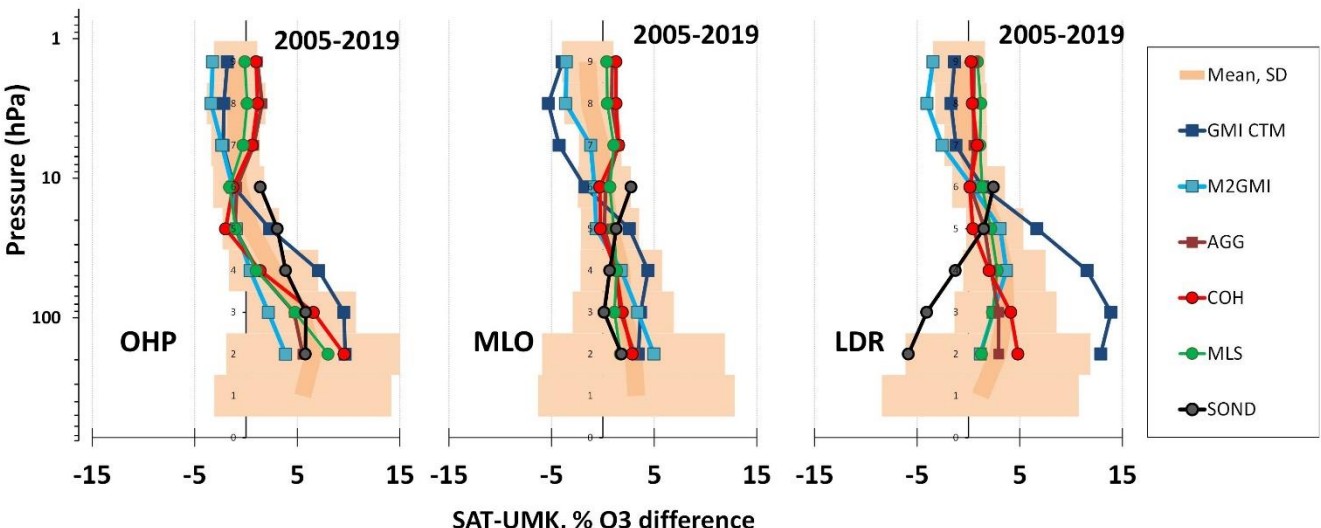

**Figure A3.** The same as the right panel of Figure 5a, but for three other Umkehr records: OHP (left), MLO (middle) and Lauder (right).

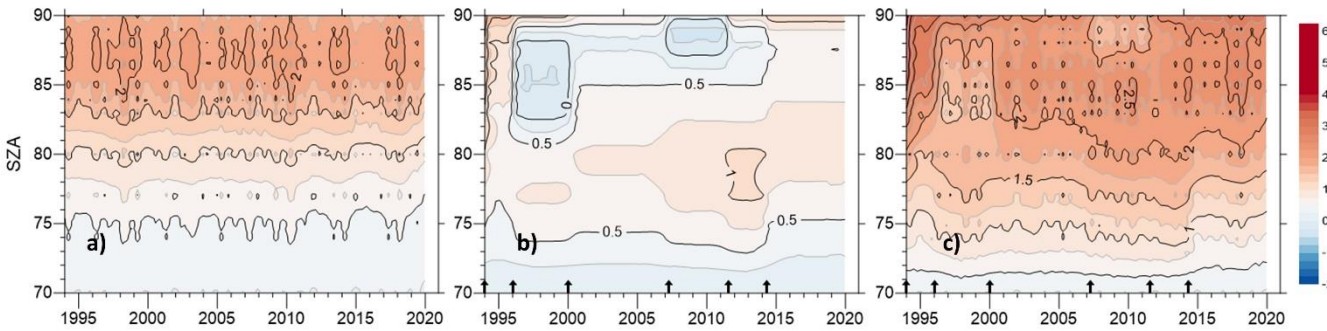

**Figure A4.** Same as Fig. 3, but for OHP.

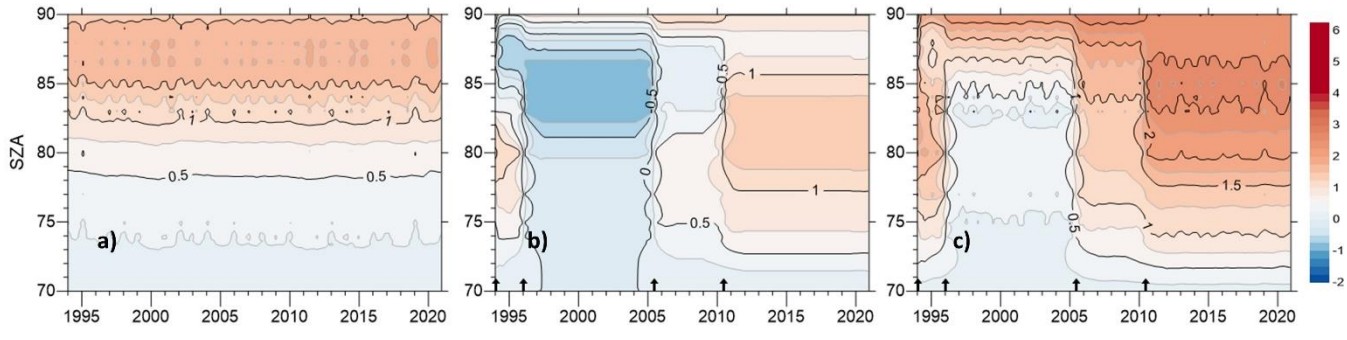

**Figure A5.** Same as Fig. 3, but for MLO.





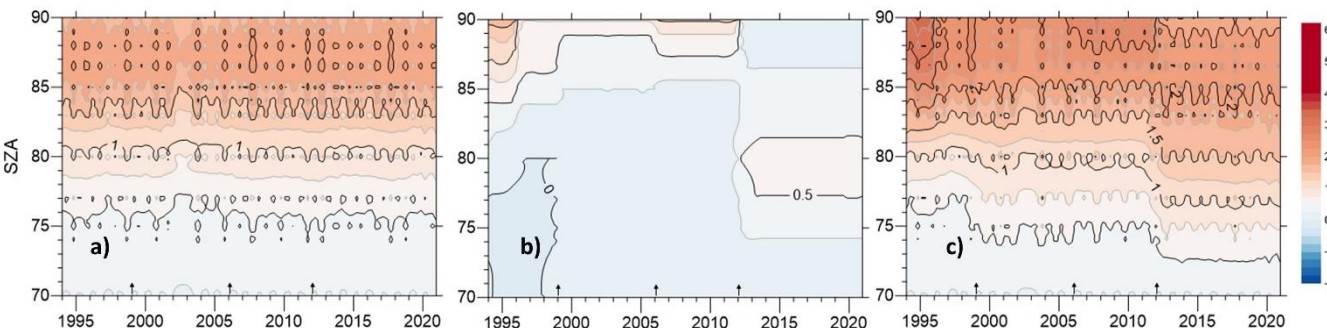

Figure A6. Same as Fig. 3, but for Lauder.






Figure A7. Same as Fig. 7, but for the OHP record.



**Figure A8. Same as Fig, 7, but for the MLO record.**







**Figure A9. Same as Fig, 7, but for the Lauder record.**





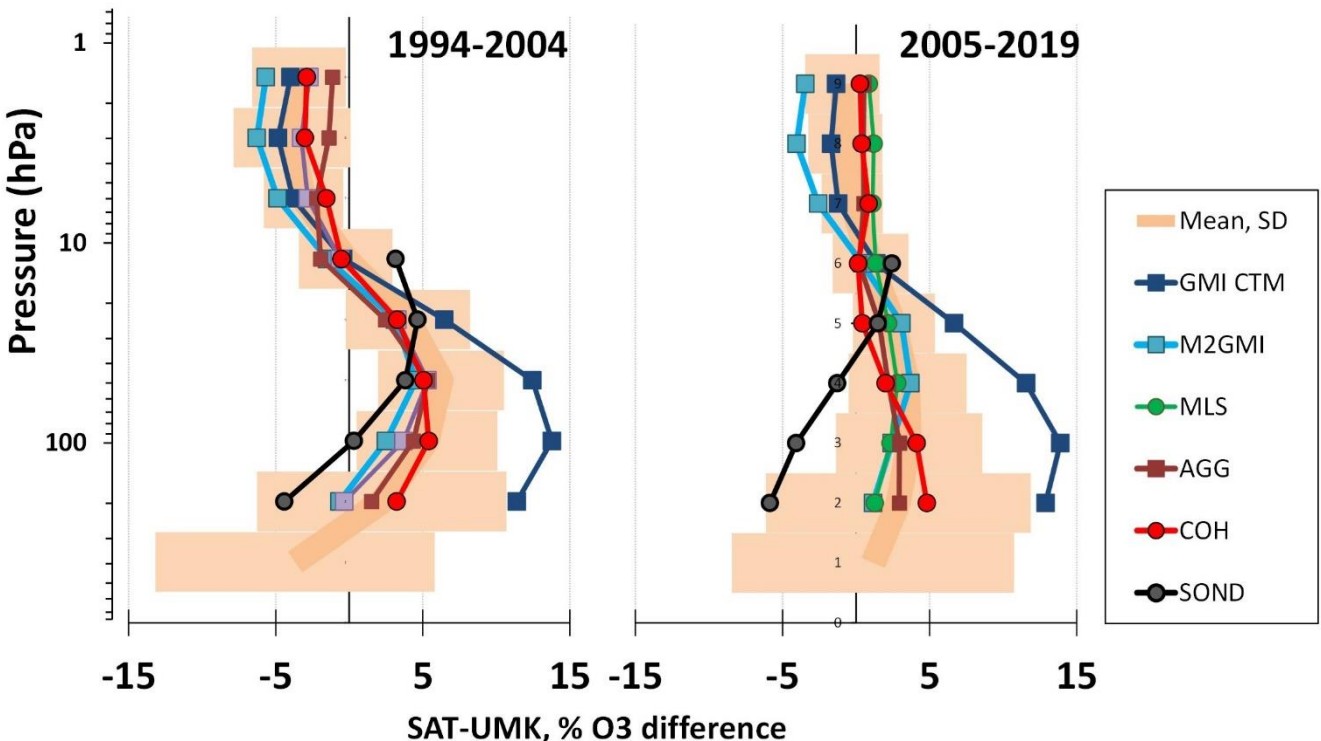

**Figure A10. Same as Figure 5a, but for the Lauder station.**



**Appendix B.**

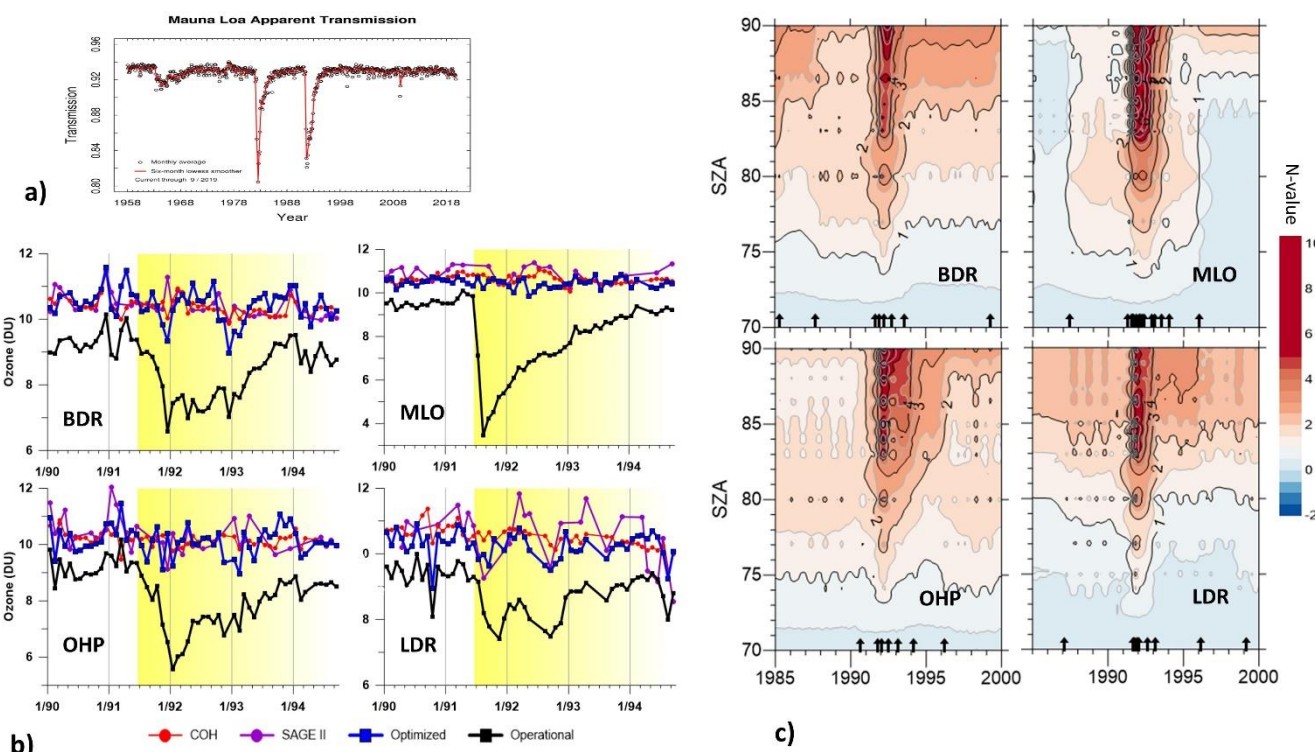

Figure B1. Correction to Boulder, OHP, MLO, and Lauder Umkehr record during the Pinatubo eruption aerosol interference. a)
MLO transmission record. (https://www.esrl.noaa.gov/gmd/grad/mloapt.html) , b) Ozone in layer 8 (DU) is shown for Umkehr
operational (black line), Umkehr optimized (blue), COH (red) and SAGE II (purple) records. The panels show Umkehr records at
Boulder, MLO, OHP and Lauder. The yellow shaded area indicates the period of Pinatubo volcanic aerosol load in stratosphere
that creates errors in operational Umkehr retrieval (large deviation in black line in 1991 and slow reduction until 1994).  c) same as
Figure 8c, but during the Pinatubo eruption. Errors indicate dates of repeated Dobson calibrations.



**Figure B2.** The time-series of corrections applied to operation Umkehr observations in Boulder, CO. a) Standardized stray light corrections (SLC) are shown at ten nominal SZAs. b) Same as panel a) but for optimized stray light corrections. Larger corrections are applied during the enhanced atmospheric aerosol loads following the El Chichon (1982) and Pinatubo (1991) volcanic eruptions. Black arrows at the bottom of the plot indicate dates of Dobson calibrations, while green arrows show when instrument replacements were done in the early part of the record). c) Final optimization that is comprised of SLC and optimized corrections.





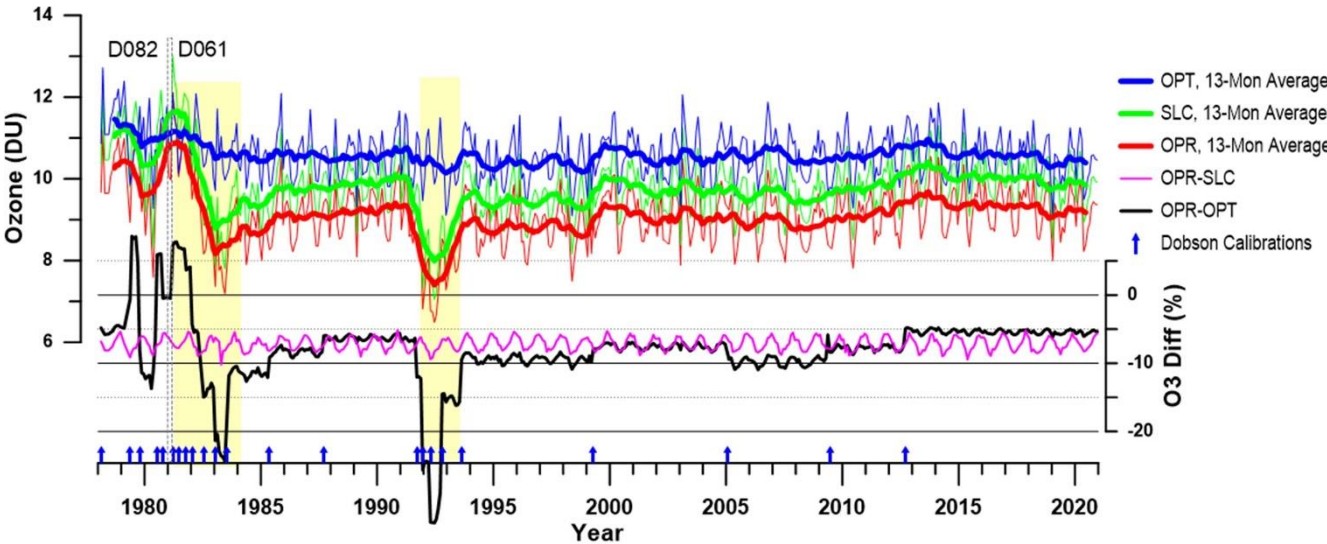

**Figure B3.** The long-term time series of Umkehr ozone time series derived in Umkehr layer 8. Upper panel: Three versions of Umkehr monthly mean ozone time series (think lines) are compared: operational (red), standard stray light correction (green) and optimized (blue). The 13-month running average (thick lines) is added to highlight the main features. Lower panel: difference between the SLC (purple line) or optimized (black line) data relative to the operational version is shown. The blue arrows indicate dates of calibrations and instrument replacements (see caption of Figure B2). The yellow areas highlight the period of errors in operational and SLC version associated with enhanced volcanic aerosol interferences (1982 El Chicon and 1991 Pinatubo eruptions) in Umkehr ozone profile retrievals.

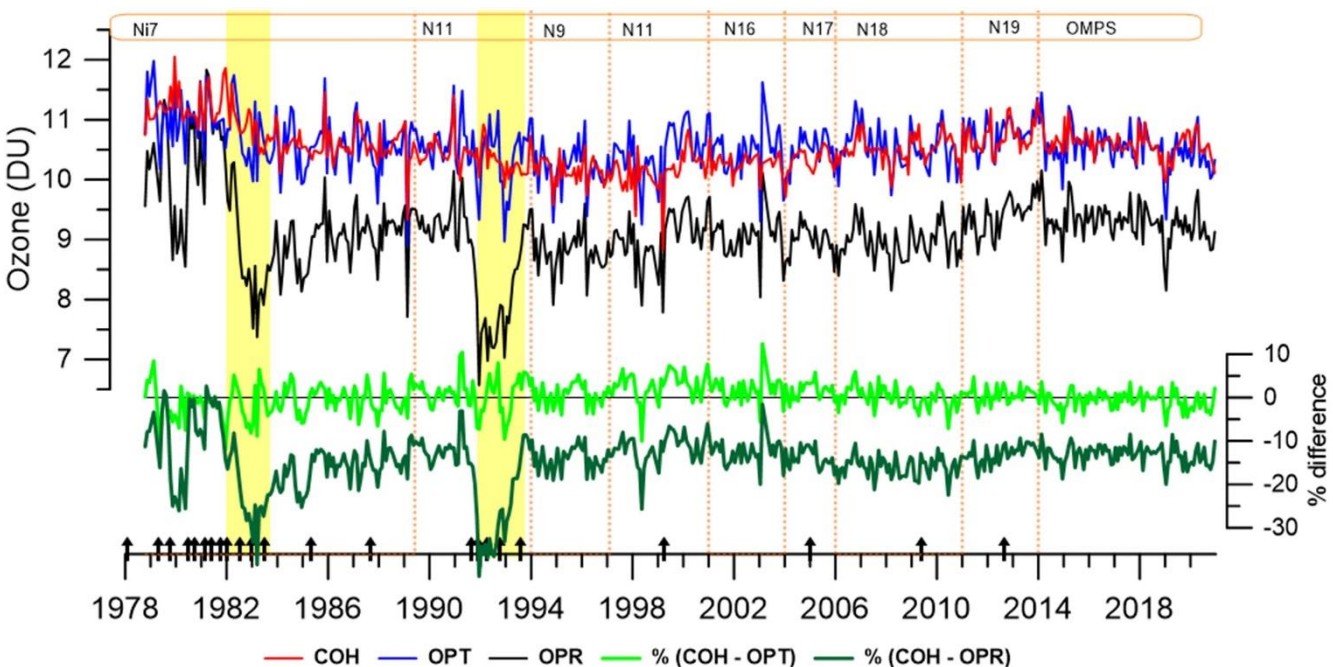





**Figure B4. The same as Fig. 7, but time series is extended to the years prior to 2000 and includes comparisons with COH observations during the volcanic time periods.**

**Appendix C**

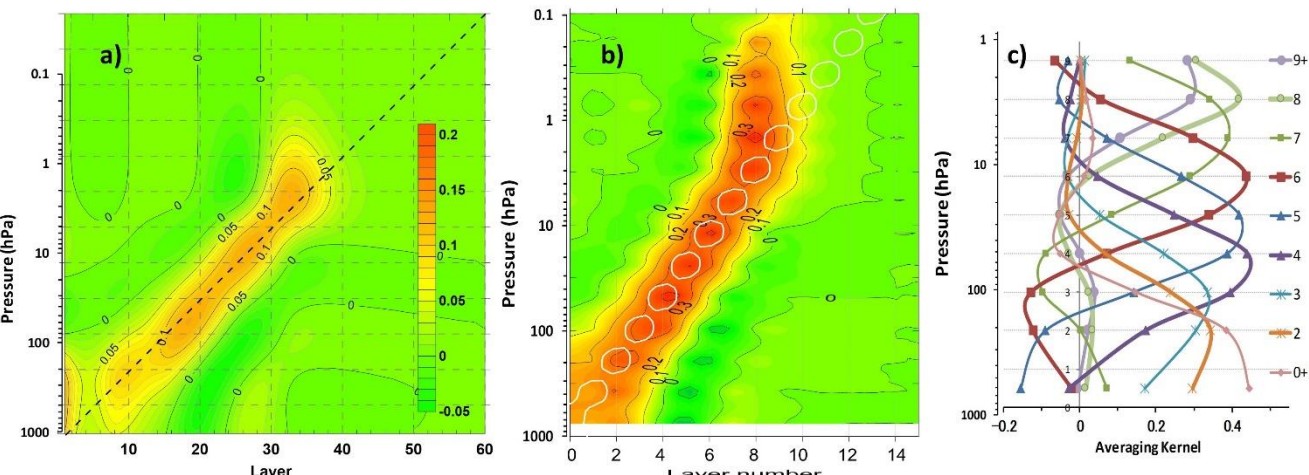

1170

**Figure C1. Averaging Kernel (AK) is shown for Umkehr retrieval in layers with a) fine vertical resolution (forward model grid, pressure on the y axes is matched to the sub-layer number, roughly every four sub-layers form one nominal Umkehr layer) b) with 16-layers and white circle shows a perfect line and c) coarse vertical resolution (optimized for independent information content in 10 nominal Umkehr layers). Colors in panel a) represent the weights of AK for each layer, where orange is the maximum (typically**

1175 **centered at the diagonal) and green color is the minimum (can be a small negative number within the uncertainty of the a priori). Colors in Panel c correspond to nominal Umkehr layers, which are shown in the legend (See Table 1). The AKs are for the operational (UMK04) retrieval of Umkehr measurements taken in Boulder on March 2, 2018.**



**Appendix D.**

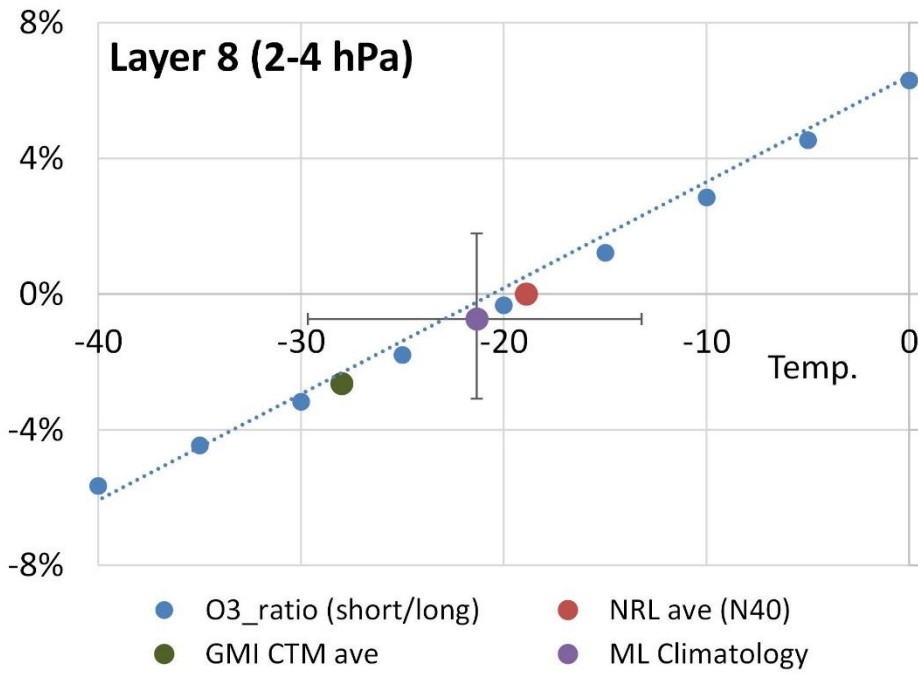

**Figure D1. Change in Umkehr layer 8 ozone as function of temperature estimated from Equation B. The blue dots represent the range of middle latitude temperature variability, the grey circle shows mean effective temperature based on 2005-2018 GMI-MERRA2 over Boulder, CO (GMIave), the red circle is the mean 35-46 N NRL temperature climatology (NRLav (N40)). The dark purple circle is the mean climatological temperature at 40-45 N (McPeter and Labow. 2011) and whiskers represent a range of seasonal variability in temperature and ozone over Boulder, CO at 4-2 hPa layer and during 2005-2018 period.**



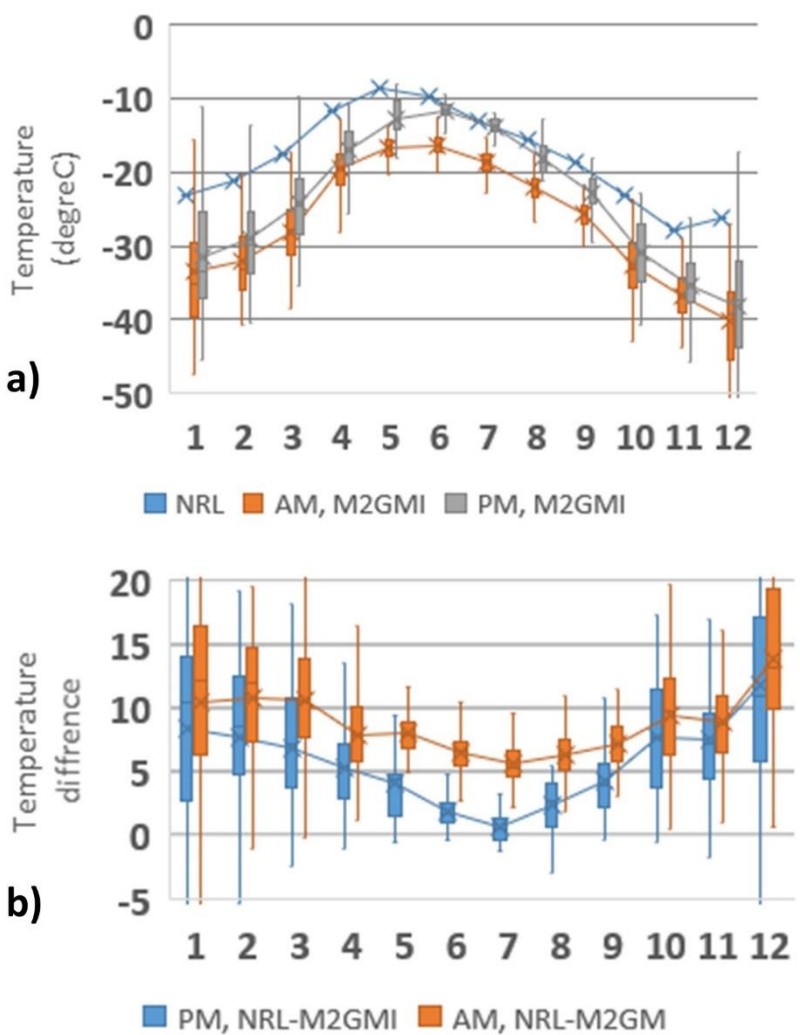

**Figure D2. a) Seasonal difference between NRL and M2GMI monthly averaged over 2010-2017 time period and selected at 15 UTC and at 2.8 hPa pressure level (Umkehr layer 8). b) the same as a), but time is at 3 UTC.**



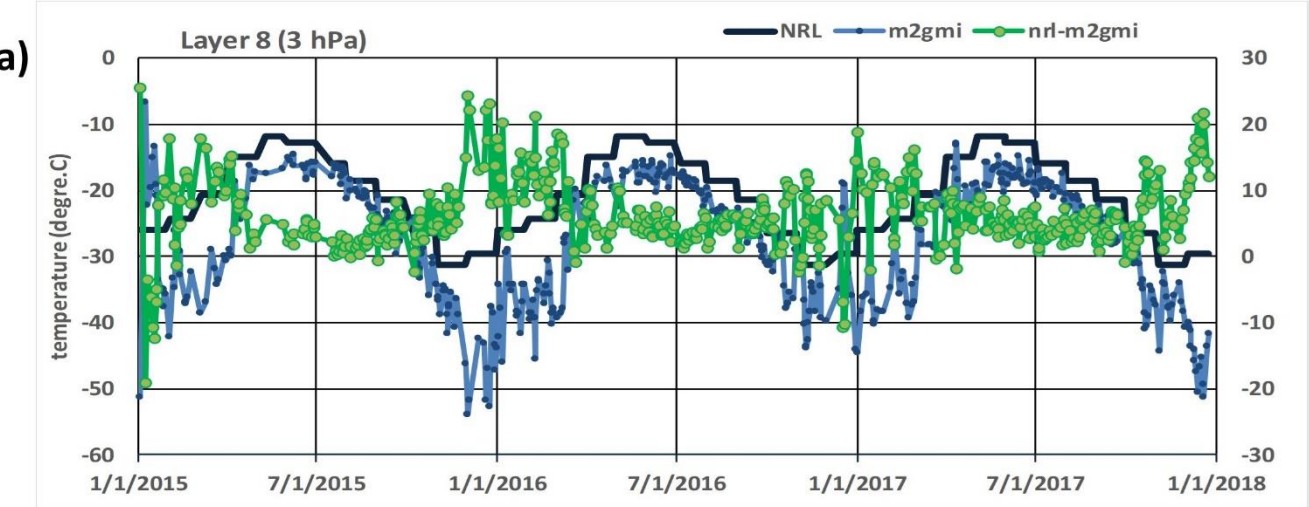

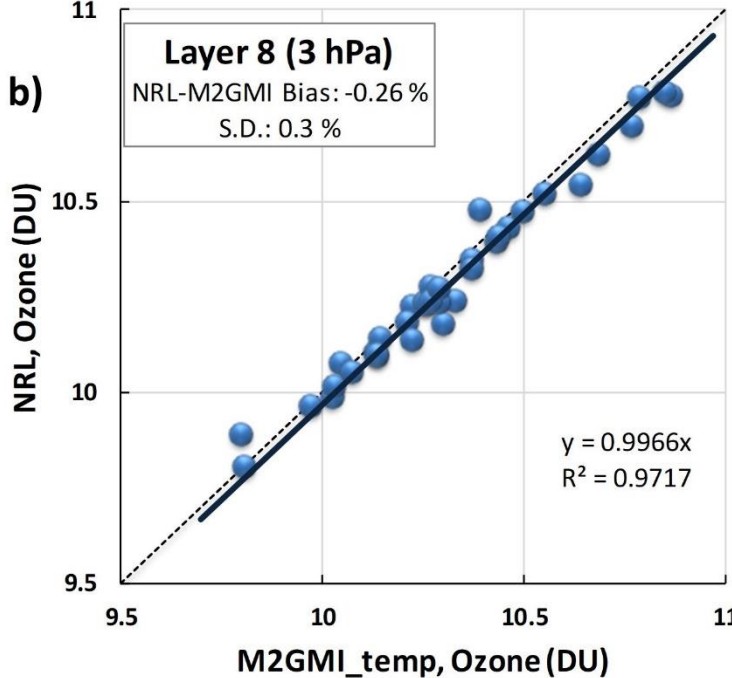

**Figure D3. Impact from the NRL climatology and M2GMI temperature on ozone variability in Umkehr layer 8 (3 hPa) at Boulder station. a) Time series of temperature. b) Ozone derived with temperature correction based on the NRL and M2GMI data. A dashed line is 1:1 slope, a solid line is the linear fit. The bias and standard deviation are shown in the box in the upper left corner, the slope and correlation coefficient are shown in the lower right corner.**