# Peer review of "Optimized Umkehr profile algorithm for ozone trend analyses."

_Atmospheric Measurement Techniques, 2021_

## Referee Comment (RC1)

**1    General Comments**

The authors describe a new method to homogenise Dobson Umkehr data recorded on long time periods. These series are based on different instruments with specific optical proprieties that are regularly calibrated and maintained. These calibration procedures which are necessary to assure good data quality and homogeneity, could introduce breaks in the data sets. The results presented in this paper show that these breaks can be adequately corrected with an advanced data reprocessing. Since the model data underlying this process (M2GMI) is available globally, the procedure could be applied to other Umkehr records eventhough the implementation of the method is certainly not strait-forward. The numerous tests and comparisons made during the development of the breaks correction process prove the great attention given to the details. The bias correction is important to reconcile the ozone profiles recorded by very different observing instruments and techniques.

The manuscript is well written, fits well within the scope of AMT so I recommend the publication after the technical corrections described below.

**2    Specific Comments**

The paper being quite long, appendix A could be put as a separate supplementary file.

P.3, L.101: the ozonesondes much finer vertical resolution is mentioned but the resolution of the Umkehr profiles is not explicitly mentioned in the paper. This information could be useful to interpret the results.

P.4, L.123: similarly for the 5% uncertainty quoted for the stratosphere. Is this value changing largely at the various Umkehr levels ? Listing the uncertainties and the vertical resolution for example in table D1 is suggested.

P.4, L.126-129: the message of the sentence "The main .... analysis." is not clear. Are new GB observations type required and/or noise reduction in existing data sets ?

P.5, L.153-155: the criterion for valid Umkehr profiles used in this study are not clear here. Are the quality flags referring to the number of iterations, the RMSE or something else ? Are the checks toward climatology applied on the raw N-values or on the final ozone values and at each levels separately ? The typical number of valid profiles per month or year used in the study would be a valuable information to mention.

P.11, L.341-343: this paragraph should be improved. It is not convincing regarding the optical characterisation of the Dobson instruments. The laser measurements probably refer to ATMOZ project measurements (cited in line 347) but are NOT used in the two mentioned references. The two lamps method is used to define the R to N conversion table that is the wedges response. The

changes of the N tables is done also with the regular standard lamps tests.

P14, L.432: first mention of table 2. It is surprising to see that the calibration listed are until 2012 / 2014 while the data are analysed until 2020. The recommended maintenance/calibrations cycle of 4-6 years is not respected. Have the calibrations of the different instruments been updated recently since biases at the end of a record are especially critical for trend analysis ?

P15, L.473: the procedure adopted for this "iterative modification" is not described. A short description would be helpful.

**3   Technical corrections / typing errors, etc**

Page 2:

- line 42: "Ball et al.,2019", 2019a or 2019b ?

- line 48: "Krzycin, J. W. and Rajewska-Wich" 2009 or 2007

- line 53: "Damadeo et al." 2018 or 2014

- line 63: "Orbe et al." 2017 or 2020

- line 63: "Dietmüller et al." 2020 or 2021

- line 65: "Morgenstein et al." not in the reference list

Page 3:

- line 70: "Petropavlovskikh et al.", 2005a, b or c

- line 77: "Staehelin, 2017" not in the reference list

- line 81: ... Vigroux *ozone* absorption ...

- line 88: "DeLuisi(1996)" not in the reference list

I stopped the references check here since there are too many mistakes. The authors are invited to go systematically through the list.

Page 4:

- line 119 : remove the extra ")"

Page 5:

- line 164 : the method is not network specific. So "... homogenization methods for *ozonesondes that were applied to* NOAA and SHADOZ networks. ..."

Page 7:

- line 217: change "," to "."

- line 221: change "htps" to "https"

- line 226: what does mean "12 pressure levels per decade" ?

Page 8:

- line 247: change 2.69×1020 to 2.69×10$^{20}$

Page 12:

- line 367: first mention of a figure of this type. The quality of these figures should be improved for clarity. The grid lines are too faint and should not be behind the shading. The layer numbers on the axis are too small. The averaged profile is almost indistinguishable and extend to layer 1 where not data are present. The common x-axis scale is nice but not large enough (e.g in Fig. A1 and A2). SAGE is not mentioned in the legend.

- line 373: add SAGE

- line 375-376: inconstancy for layer 6.

- line 377: for layer 1, the individual differences are not displayed. Is there a reason or just too large differences ?

- line 381 + 383-384 : the discussion of GMI-CTM and M2GMI differences differs from what is seen in the figures !?

- line 384: on figure 1a), there are also 5 data sets !

Page 14:

- line 432: first mention of the Table 2. The table is not restricted to Boulder data as mentioned in the header. The sentence starting with CAL is not clear, CAL being not defined. "Updated WD" is supposedly a change of WinDobson version. What can be expected from that change ?

- line 446-448: redundant sentences

- line 446: first mention of fig 2 a). Here also the "Mean" profile is indiscernible.

- line 454: On figure 2b), the value at SZA=90 is 1.1 while on fig 1a) it is closer to 1.3. Should these two number match ?

Page 15:

- line 459-460: redundant sentences

- Line 478: first mention of fig 3. No instrument replacement occur on this time period so remove the last words of the table header.

Page 16:

- line 491: the offsets mentioned are for the "OPT - SLC" differences

- line 502: first mention of fig B3. The legend and caption of fig B3 for the black line is SLC-OPT not OPR-OPT which would not have a sign change at the beginning in the 80s.

- line 513: first mention of fig 5. remove "(old 6)" in the caption

Page 17:

- line 524-525: the dark-blue and light-blue curve show the other way around, M2GMI is close to zero

Page 18:

- line 567: ... with a mean *positive* bias ...

- line 578: ... (see Fig 5a.) not 6a

Page 19:

- line 596: first mention of fig A1, A2 and A3. The time period mentioned is "2005-2019" contrarily to table 1 time periods and fig 1

- line 605: MLO and OHP are reversed ?

Page 21:

- line 654: panel b

- line 660: (*purple* line)

- line 661: ... for *four* stations, add OHP in the list

- line 666: the rest of Appendix B is not finished and Figures B2, B3 and B4 not mentioned !!!!!! Either complete or remove this part

- line 669: first mention of Table C1. Are the 21 layers column used anywhere in this study ? They are not referred on line 671. Correct the table header for "... top *of* the ..."

Page 22:

- line 712: Figure *D1*. In the caption, reference to an equation B that is not present !!

Page 23: correct all the figures reference that should be D-something as well as the y-axis label of figure D2b "Temperature difference"

---

## Author Comment (AC1)

The authors would like to thank referees for the time and effort to review this paper. It helped us to revise the manuscript to make it clearer and more concise. We have addressed all the referee comments as described below. The referee comments are shown in black and our responses are shown in blue (also italic).

**Reviewer 1**

General Comments

The authors describe a new method to homogenize Dobson Umkehr data recorded on long time periods. These series are based on different instruments with specific optical proprieties that are regularly calibrated and maintained. These calibration procedures which are necessary to assure good data quality and homogeneity, could introduce breaks in the data sets. The results presented in this paper show that these breaks can be adequately corrected with an advanced data reprocessing. Since the model data underlying this process (M2GMI) is available globally, the procedure could be applied to other Umkehr records even though the implementation of the method is certainly not straight-forward. The numerous tests and comparisons made during the development of the break's correction process prove the great attention given to the details. The bias correction is important to reconcile the ozone profiles recorded by very different observing instruments and techniques.

The manuscript is well written, fits well within the scope of AMT so I recommend the publication after the technical corrections described below.

*Authors would like to thank the reviewer for all the comments.*

2 Specific Comments

The paper being quite long, appendix A could be put as a separate supplementary file.

*Authors agree that the paper appears to be long as it requires a lot of details to address specifics of Umkehr corrections due to natural (e.g., volcanic) and instrumental interferences. We agree to submit the Appendix as a supplemental file.*

P.3, L.101: the ozonesondes much finer vertical resolution is mentioned but the resolution of the Umkehr profiles is not explicitly mentioned in the paper. This information could be useful to interpret the results.

*Thank you for pointing to the lack of discussion of the vertical resolution of Umkehr profiles. We modified the sentence to include reference to Table C1 that provides information about the vertical resolution of Umkehr profiles.*

*"These profiles have relatively low vertical resolution (see Appendix C and Table C1 for further information). Umkehr records do not come as a replacement to other ground-based observations of the ozone profile, but rather serve to complement them."*

P.4, L.123: similarly for the 5% uncertainty quoted for the stratosphere. Is this value changing largely at the various Umkehr levels ? Listing the uncertainties and the vertical resolution for example in table D1 is suggested.

*Thank you for the suggestion to include Umkehr errors in Table C1. The total uncertainty for Umkehr layers is a combination of instrumental errors and smoothing errors. Petropavlovskikh et al. (2005) have a Figure 2 that summarizes the relative error of the retrieval as a function of layers. The errors due to volcanic aerosols (systematic errors) are also summarized in Table 1. We added the reference to Petropavlovskikh et al. 2005 paper.*

[Figure]

**Figure 2.** Relative error of the retrieval as function of layers: standard deviation in monthly averaged ozone divided by mean ozone. Effect of minimum SZA is shown by three lines representing three scenarios (60, 70 and 77 degree normalization SZA).

*The vertical resolution for the Umkehr profile is shown in Figure 3 (below, the orange line marked as Dobson/Brewer) of Hassler et al. 2014 publication. Hassler's paper is also now referenced.*

[Figure]

P.4, L.126-129: the message of the sentence "The main .... analysis." is not clear. Are new GB observations type required and/or noise reduction in existing data sets ?

*Thank you for pointing to unclear meaning of this sentence. We updated the text.*

*"The main objective of this paper is to reduce noise in the existing Umkehr records and therefore improve its suitability for detection of relatively small trends (e.g. 1-3 % over 2000-2016 period, LOTUS, 2019). In addition, continuous improvement of the satellite retrieval algorithms requires ground-based observations of high accuracy and stability, which optimized Umkehr record aims to provide."*

P.5, L.153-155: the criterion for valid Umkehr profiles used in this study are not clear here. Are the quality fags referring to the number of iterations, the RMSE or something else ? Are the checks toward climatology applied on the raw N-values or on the final ozone values and at each levels separately ? The typical number of valid profiles per month or year used in the study would be a valuable information to mention.

*Thank you for the question. We added requested information*

*"The quality check of the retrieved ozone profile includes assessment of the number of iterations (fewer than 4 is considered a good profile) and the condition that the difference between observed and retrieved N-values at all*

*SZAs remains within measurement uncertainty (Petropavlovskikh et al., 2005). The number of accepted Umkehr profiles per month depends on the station geo-location and season and can vary between a few (e.g. at Boulder in spring due to seasonal increase in clouds) and up to 60 (e.g. at MLO when counting both morning and afternoon retrievals in winter), but on average Umkehr stations observe 15 profiles per month or more (~30 profiles at MLO)."*

P.11, L.341-343: this paragraph should be improved. It is not convincing regarding the optical characterization of the Dobson instruments. The laser measurements probably refer to ATMOZ project measurements (cited in line 347) but are NOT used in the two mentioned references. The two lamps method is used to define the R to N conversion table that is the wedges response. The changes of the N tables is done also with the regular standard lamps tests.

*Thank you for pointing out the incompleteness of the information in this paragraph and the use of wrong references. We made changes to the text.*

*"The measurement of a Dobson slit function is not a simple task. The original method used a model 783 McPherson spectrophotometer to determine the slit functions for Dobson 083 (Komhyr et al, 1993). The method restricted the slit function to the core band-pass and did not provide information about out-of-band light rejection."*

P14, L.432: First mention of table 2. It is surprising to see that the calibration listed are until 2012 / 2014 while the data are analyzed until 2020. The recommended maintenance/calibrations cycle of 4-6 years is not respected. Have the calibrations of the different instruments been updated recently since biases at the end of a record are especially critical for trend analysis?

*Thank you for this question. We checked the entire record (through 2020) for step changes during calibration activities and found that Dobson instruments are calibrated every 4-6 years. However, not all calibration activities create the step change, or at least the method does not allow to discern changes that are less than measurement noise). Another limitation of this method is that it requires to have at least 3 years after the calibration to compare observations before and after the calibration date. Therefore, if calibration happens within the last three years of the record, the optimization method is not capable to detect the step change. However, MLR trend analyses of the 20-year long record (2000-2020) should not have a statistically significant impact from the potential bias in the last 2 years of the record. In fact, after submission of paper we noticed that calibration at Lauder in 2017 has created a step change in the record. We reprocessed the Lauder data and updated the Figures and sections in the paper that discuss the optimization of the Lauder record.*

*We added the following text*

*"These dates do not represent the calibration record of station instrument. Not all calibration activities create a step change in Umkehr records. Alternatively, the optimization method does not allow one to discern changes that are less than measurement noise. Another limitation of this method is that it requires at least 3 years of the record after the calibration to derive a correction. Therefore, if calibration happens within the last two years of the record, the optimization method is not capable to detect the step change until a longer period becomes available."*

*We also removed "Dobson calibration history" from the Table 2 title to avoid a confusion.*

P15, L.473: the procedure adopted for this "iterative modification" is not described. A short description would be helpful.

*We added information to explain the iterative process.*

*"Therefore, the step-by-step adjustment of the M2GMI-based correction curve is performed using 0.1 N-value increments at one SZA at a time. The adjusted correction is tested for the Umkehr retrieval. The iterative process continues until the remaining bias between optimized Umkehr (M2GMI\*) and MLS ozone profiles (Fig.2 c, dark line) is minimized in the 2005-2018 period. The final N-value adjustment for the 2005-2018 period is marked as M2GMI\* (black line in Fig.2 a). This additional correction to M2GMI optimization is applied to all M2GMI empirical corrections prior to re-processing the entire Umkehr record."*

Technical corrections / typing errors, etc
Page 2:
• line 42: "Ball et al.,2019", 2019a or 2019b ?
We changed reference to Ball et al., 2019a.

• line 48: "Krzycin, J. W. and Rajewska-Wich" 2009 or 2007
*Thank you for noticing inconsistencies in references (here and below) . We changed 2009 to 2007.*

• line 53: "Damadeo et al." 2018 or 2014
*We changed 2018 to 2014.*

• line 63: "Orbe et al." 2017 or 2020
*We changed 2020 to 2017.*

• line 63: "Dietm•uller et al." 2020 or 2021
*This paper is now published, thus 2021 is correct. We updated paper in the reference section:*
*Dietmüller, S., Garny, H., Eichinger, R., and Ball, W. T.: Analysis of recent lower-stratospheric ozone trends in chemistry climate models, Atmos. Chem. Phys., 21, 6811–6837, https://doi.org/10.5194/acp-21-6811-2021, 2021.*

• line 65: "Morgenstein et al." not in the reference list
*We included this reference and changed 2018 to 2017 in the txt*
*Morgenstern, O., Hegglin, M. I., Rozanov, E., O'Connor, F. M., Abraham, N. L., Akiyoshi, H., Archibald, A. T., Bekki, S., Butchart, N., Chipperfield, M. P., Deushi, M., Dhomse, S. S., Garcia, R. R., Hardiman, S. C., Horowitz, L. W., Jöckel, P., Josse, B., Kinnison, D., Lin, M., Mancini, E., Manyin, M. E., Marchand, M., Marécal, V., Michou, M., Oman, L. D., Pitari, G., Plummer, D. A., Revell, L. E., Saint-Martin, D., Schofield, R., Stenke, A., Stone, K., Sudo, K., Tanaka, T. Y., Tilmes, S., Yamashita, Y., Yoshida, K., and Zeng, G.: Review of the global models used within phase 1 of the Chemistry–Climate Model Initiative (CCMI), Geosci. Model Dev., 10, 639–671, https://doi.org/10.5194/gmd-10-639-2017, 2017.*

Page 3:
• line 70: "Petropavlovskikh et al.", 2005a, b or c
*Good point, there are several publications in 2005. We corrected the reference to Petropavlovskikh et al., 2005a.*

• line 77: "Staehelin, 2017" not in the reference list
*We added the reference to the paper and updated the text to "Staehelin et al., 2018).*
*Staehelin, J., Viatte, P., Stübi, R., Tummon, F., and Peter, T.: Stratospheric ozone measurements at Arosa (Switzerland): history and scientific relevance, Atmos. Chem. Phys., 18, 6567–6584, https://doi.org/10.5194/acp-18-6567-2018, 2018.*

• line 81: ... Vigroux ozone absorption ...
*We added "ozone" to the text*

• line 88: "DeLuisi(1996)" not in the reference list
I stopped the references check here since there are too many mistakes. The authors are invited to go systematically through the list.
*Thank you for catching the inconsistencies in the references. We checked all references and made corrections.*

Page 4:
• line 119 : remove the extra ")"
*Removed*

Page 5:
• line 164 : the method is not network specific. So "... homogenization methods for ozonesondes that were applied to NOAA and SHADOZ networks...."
*Agree, the sentence was corrected*

Page 7:

• line 217: change "," to "."
*Corrected*

• line 221: change "htps" to "https"
*Corrected*

• line 226: what does mean "12 pressure levels per decade" ?
*Retrieved ozone values are computed for 12 levels per decade change in pressure. That is, 12 levels are selected between 1 and 10 hPa, another 12 levels are selected between 10 and 100 hPa, and also between 100 and 1000 hPa. The spacing is linear in log pressure. This information is provided in Livesey et al. (2020).*
*We added this information in the section that describes MLS observations.*

Page 8:
• line 247: change 2.69×1020 to 2.69×1020
*Corrected to $2.69x10^{20}$*

Page 12:
• line 367: First mention of a Figure of this type. The quality of these figures should be improved for clarity. The grid lines are too faint and should not be behind the shading. The layer numbers on the axis are too small. The averaged profile is almost indistinguishable and extend to layer 1 where not data are present. The common x-axis scale is nice but not large enough (e.g. in Fig. A1 and A2). SAGE is not mentioned in the legend.
*Thank you for comments about Figure quality. We improved this type of Figures in the new version. We included SAGE II in the Figure caption.*

• line 373: add SAGE
*We added SAGE II to the text.*

• line 375-376: inconstancy for layer 6.
*We replace layer 6 with 4 in the sentence that lists layers with bias larger than 5 %.*

• line 377: for layer 1, the individual differences are not displayed. Is there a reason or just too large differences ?
*Results for layer 1 were not included because SAGE II or MLS satellite records do not have consistent ozone information below 250 hPa, whereas vertical grid of the SBUV/OMPS profiles is coarse for interpolation.*
*(ATTN. Jeannette or Larry – should I remove comparisons with SBUV/OMPS in layer 2, which is between 250 and 125 hPa due to insufficient resolution of the output grid).*
*Irina, It is up to you. The comparisons give more information on the SBUV/OMPS than the Umkehr as the ground-based sees the full column change better than the space-based.*

• line 381 + 383-384 : the discussion of GMI-CTM and M2GMI differences differs from what is seen in the figures !?
*You are correct. The description of results for GMI CTM and M2GMI were switched in the text.*
*We changed the description in the text to:*
*"The M2GMI model comparisons show the smallest biases, except for the largest negative bias in layer 2 found in both time periods and an increased positive bias in layer 5 in 2005-2020 period. The GMI CTM biases in layers 6-9 are similar in magnitude to the M2GMI biases, whereas they grow larger in layers 3-5.  Moreover, in 2005-2020 period (right panel) both M2GMI and GMI CTM biases in layers 3-5 increase relative to 1994-2004 (left panel) comparisons, and GMI CTM biases becomes the largest positive biases among all datasets."*

• line 384: on Figure 1a), there are also 5 data sets !
*We changed the statement.*
*"The mean offset is calculated by averaging results from all datasets except SAGE II (six datasets before and after 2005)"*

Page 14:

• line 432: First mention of the Table 2. The table is not restricted to Boulder data as mentioned in the header. The sentence starting with CAL is not clear, CAL being not defined. "Updated WD" is supposedly a change of WinDobson version. What can be expected from that change ?

*Thank you for noticing missing stations listed in Table 2.*
*We made a correction.*
*"Table 2 contains the dates and time periods selected to apply empirically derived adjustments to Dobson observations in Boulder, CO, OHP, France, Mauna Loa, Hawaii, and Lauder, New Zealand."*
*We also updated the text in Table 2. We removed the sentence that started with "CAL". We added explanation about the Updated WD.*
*"Table 2. Umkehr N-value optimized corrections for each nominal solar zenith angle (70° - 90° SZA) are shown for four Umkehr stations. All corrections are normalized to 70° SZA (set to zero at 70° SZA). The correction period is between the dates indicated in the second column. The last correction is through the end of 2020. The "Updated WD" note on the right of the table identifies the date when the WinDobson system was installed for Dobson automation."*

*The WinDobson system changes are described in lines 150-153. We modified text to explain changes to data processing.*
*"During the automation, the observational process (i.e. frequency of observations, signal-to-noise, cloud clearance, etc.) is changed. The software uses the near-IR cloud detector to screen the Umkehr data for clear sky conditions, interpolates screened observations to 12 nominal SZAs, adds total column ozone information, processes data and checks retrieved ozone profiles for quality flags and against station climatological variability (+/-2 standard deviations). This process results in the improved quality assurance of observations and reduces cloud-induced anomalies in the Umkehr data.*

• line 446-448: redundant sentences
*We did not find redundant sentences in our version of the text. Perhaps, it was a mistake in the pdf version the reviewer might have received.*

• line 446: First mention of Fig 2 a). Here also the "Mean" profile is indiscernible.
*The mean profile line was added to the plot in Fig 2 a).*

• line 454: On Figure 2b), the value at SZA=90 is 1.1 while on Fig 1a) it is closer to 1.3. Should these two numbers match ?
*Panel a) shows mean values, whereas panel b show histogram. Therefore, the maximum at 1.1 N-value in histogram distribution might not be the same as mean value.*

Page 15:
• line 459-460: redundant sentences
*We did not find redundant sentences in our version of the text. Perhaps, it was a mistake in the pdf version the reviewer might have received.*

• Line 478: First mention of Fig 3. No instrument replacement occurs on this time period so remove the last words of the table header.
The header of Table 2 was updated.
*"Table 2. Umkehr N-value optimized corrections for each nominal solar zenith angle (70° - 90° SZA) are shown for four Umkehr stations. All corrections are normalized to 70° SZA (set to zero at 70°). The correction period is between the dates indicated in the second column. The last correction is through the end of 2020. The "Updated WD" note on the right of the table identifies the date when the WinDobson system was installed for Dobson automation.*

Page 16:

• line 491: the offsets mentioned are for the "OPT - SLC" differences
*Thank you for catching the error. We replaced OPR with OPT in the sentence.*

• line 502: First mention of Fig B3. The legend and caption of Fig B3 for the black line is SLC-OPT not OPR-OPT which would not have a sign change at the beginning in the 80s.
*Thank you for noticing the error. The Figure B3 is now removed.*

• line 513: First mention of Fig 5. remove "(old 6)" in the caption
*Correction was made*

Page 17:
• line 524-525: the dark-blue and light-blue curve show the other way around, M2GMI is close to zero
We corrected this error
*"Also positive biases (up to 8 %) are found in comparisons with GMI CTM profiles in 2005-2020 period, whereas the bias is negligible in M2GMI comparisons."*

Page 18:
• line 567: ... with a mean positive bias ...

*Correction was made*

• line 578: ... (see Fig 5a.) not 6a
*Thank you for noticing discrepancies in numbering of Figures. The correction was made.*

Page 19:
• line 596: First mention of Fig A1, A2 and A3. The time period mentioned is "2005-2019" contrarily to table 1 time periods and Fig 1
*We made correction; it is 2005-2020 now.*

• line 605: MLO and OHP are reversed ?
*Not clear what the reviewer means here. Are you asking about mean bias in layer 2? I believe it is described correctly – less than 5 % at MLO and about  5 % at OHP.*

Page 21:
• line 654: panel b
*This was corrected in the final text*

• line 660: (purple line)
*This was corrected in the final text*

• line 661: ... for four stations, add OHP in the list
*This was corrected in the final text, OHP was added to the list.*

• line 666: the rest of Appendix B is not finished and Figures B2, B3 and B4 not mentioned !!!!!! Either complete or remove this part
*Thank you for noticing the missing text. We decided not to include Figures B2-B4.*

• line 669: First mention of Table C1. Are the 21 layers column used anywhere in this study ? They are not referred on line 671. Correct the table header for "... top of the ..."
*The 21 layers are the standard output for COH products. For comparisons between Umkehr and these datasets we first created partial columns by integrating ozone from 21 COH layers. Next, we did interpolation to Umkehr pressure levels in log-pressure space. In the last step we used a difference between partial columns to convert COH profiles to Umkehr layers.*

*We updated the Header for Table C1*
*"Table C1. Umkehr and COH pressure layer grids. The layer is defined by the pressure at the bottom of the layer. The pressure at the top layer is between the pressure level and the top of the atmosphere.* ==The standard Umkehr output is in 10-layer system (2 leftmost columns). Column 3 provides estimate of the total uncertainty in 10 Umkehr== *layers (e.g. results from Figure 2 in Petropavlovskikh et al., 2005). The 16-layer system (columns 4 and 5) is used for the AK output. The 61-layer system (columns 6 and 7) is utilized in the forward model. The 21-layer system (columns 8 and 9) is the standard COH profile output. All COH profiles are interpolated to the 10- or 61- layer Umkehr grid for intercomparisons."*

Page 22:
• line 712: Figure D1. In the caption, reference to an equation B that is not present !!
*Thank you for noticing the missing equation. We changed the D1 Figure caption.*
*"Change in Umkehr layer 8 ozone as function of temperature estimated from the second-degree polynomial fit (see coefficients in Table D1)." We also replaced 2018 with 2020 in the Figure caption text and changed GMI MERRA2 to GMI CTM to make it consistent with the rest of the paper.*

Page 23: correct all the figures reference that should be D-something as well as
the y-axis label of figure D2b "Temperature difference"
*The correct Figure references and now used in the final version of edited draft.*

---

## Author Comment (AC2)

The authors would like to thank referees for the time and effort to review this paper. It helped us to revise the manuscript to make it clearer and more concise. We have addressed all the referee comments as described below. The referee comments are shown in black and our responses are shown in blue (also italic).

Reviewer 2 comments

**General comments:**

The Authors present reprocessed, homogenized and overall improved datasets containing long-term Umkehr retrievals of Ozone profiles derived using measurements from Dobson spectrometers. The dataset is then formally compared against several satellite records while considering the Averaging Kernels of the Umkehr retrievals. The manuscript is quite long but comprehensive, and would serve as valuable reference for future works in this subject.

The implementation of the method is not trivial and probably beyond the current capacity of other groups operating Dobson spectrometers. Nevertheless, this work demonstrates the added value of these observations and their usefulness in the future. As improved instruments, algorithms and spectroscopy arise, long-term historical records such as those presented here become more important as benchmark in monitoring the history and evolution of the Ozone layer.

This manuscript fits well within the scope of AMT. Therefore, I recommend its publication after addressing the comments of Reviewer 1 and some of the comments and corrections below.

**Specific comments:**

Most of the figures are well-made, legible and contain the right amount of information. The Red and Green lines/markers can be difficult to distinguish for some readers (e.g. Fig. 1, 2, 4 ,5, etc.), but I think they are still bordering on OK in the plots where they appear. Some of the figure captions need to be checked for typographical errors.

*Thank you for your comments regarding Figure colors. We addressed your concerns with changing the pure green and red colors to more color-blind acceptable hues.*

*We also updated Figure captions and checked for errors (please see figures in the revised manuscript).*

P.9, Line 84: On the step change in the GMI CTM: I would suggest to provide at least one sentence of explanation on what this step change is, and why it happened.

*Th reviewer is probably referencing text on Line 284 on page 9.*

*We addressed reviewers comment and modified the text.*

*"The step-change in the GMI CTM ozone record in 1998 was documented (Stauffer et al, 2019 and references therein). It was a result of the introduction of microwave radiance observations from a series of Advanced Microwave Sounding Unit (AMSU) sensors into the MERRA-2 observing system (Gelaro et al. 2017). The 1998 change as well as the addition of MLS temperature assimilation in the upper stratosphere in 2004 strongly impacted the MERRA-2 dynamical fields (Gelaro et al., 2017; Long et al., 2017).*

*We also added the reference below to the list of references.*

Long, C. S., Fujiwara, M., Davis, S., Mitchell, D. M., and Wright, C. J.: Climatology and interannual variability of dynamic variables in multiple reanalyses evaluated by the SPARC Reanalysis Intercomparison Project (S-RIP), Atmos. Chem. Phys., 17, 14593–14629, https://doi.org/10.5194/acp-17-14593-2017, 2017.

P.10-11, Lines 325-325: The description of the Dobson optical system needs some revision, as already noted by Reviewer 1. Perhaps it is also worth to mention that the Dobson spectrophotometer is a double monochromator. Also important: the optical wedge attenuates the long wavelength signal, the Q-levers indicate the positions of the wavelength pairs (A, B, C or D), which depend on the temperature inside the instrument. The photomultiplier registers the alternating signals from the short wavelength, which is absorbed by Ozone, and the long wavelength attenuated by the optical wedge, resulting in a measureable current.

We appreciate comments from both reviewers with regards to the description of Dobson optical system description. At the beginning of section 3.1 we state that Dobson consists of two monochromators, thus it is a double monochromator.  Here is the modified text.

"The Dobson consists of two monochromators and a slit plate for selecting two bands (pairs) of the UV solar spectrum approximately 20 nm apart. The Q-levers indicate the position of the wavelength pairs (A, B, C or D), which also depends on the temperature inside of the instrument.  The  photomultiplier tube registers the alternating signals from the short wavelength, which is absorbed by ozone, and the long wavelength attenuated by the optical wedge, resulting in the measurable current (see Komhyr and Evans, 2006 for further details)."

Also, in Section 3.1., The Umkehr N-values need a better explanation for the non-specialist reader.

Thank you for your comment. Section 3.1 is concerned with the measurement uncertainties in Dobson system. The Umkehr measurement technique is described in the Introduction section (line 72-86).   We adjusted one of the sentences  to add N-value definition.

"The **logs of** ratio of the observed radiances **(also called N-values)** increase with increasing SZA and at about 86° SZA reverse and starts to decrease up to 90° SZA, which grants the observation its name since "Umkehr" means reversal or change in German."

P.13, Section 3.3, First Paragraph

This is a long paragraph that could be divided into two or three paragraphs for easier reading.

Thank you for suggestion. We split the paragraph into two to separate the description of the forward model simulations.

Also, the Authors mention that the Umkehr N-values are simulated for an idealized Dobson instrument. So, I would like to ask:

Please see responses below questions.

1. How far or close to ideal are the Dobsons used here?

   There is no ideal Dobson. We replaced the "idealized" with the "generic". The word "idealized" is used here to refer to the slit function published by Komhyr (1993). The experimentally determined slit functions of the Dobson 083 instrument  can be described as a triangle and trapezoidal shapes. The mapping method did not provide information about the out-of-band light rejection. Most Dobson  instruments have very similar core band-passes as discussed in Kohler et al. (2018) paper. However, the nominal slit functions do not include the information about the out-of-band contribution of the light scattered into the instrument. As written in

Section 3.1, the full mapping of the band-pass of each instrument is rarely done as the focus of the Dobson network is on the total ozone observations that are limited to SZAs where contribution of the stray light is minimal. However, Umkehrs are using observations at low SZAs where contribution of the stray light becomes significant to offset simulations in the forward model that is using only the core band-pass information. The omission to include the contribution of the out-of-band light can create a vertically distributed bias (approximately +/- 5 %) in the Umkehr retrieved ozone profile (Petropavlovskikh et al., 2011).

2.  Do the stations keep a record of the instrument characteristics mentioned in Sec. 3.1 (slit functions, response, etc.)?

    Not to my knowledge. As mentioned above, slit functions of only several Dobson instruments were mapped so far with the laser beam (Komhyr, 1993; Kohler et al, 2018). The mercury lamp tests are used monthly to assure that the slit spectral positions are not drifting away from the center of the nominal bandpass. If the drift is detected, the changes to the Dobson operations are adjusted and spectral stability of the instrument is verified during intercomparisons with the standard instrument.

3.  Would it not be useful to include a Figure of these characteristics, perhaps in the Appendix?

    We do not have a full slit function for Dobson instruments at the analyzed station. The impact of the best-guess out-of-band light contribution to Umkehr profiles retrieval errors is already discussed in Petropavlovskikh et al. (2011) and Evans et al. (2009). We decided that due to the already long appendix, it is better to provide references to the published papers and let the reader read the discussions provided in those papers.

Appendix C: Umkehr Averaging Kernels:

It would be interesting to know the Degrees of Freedom for Signal (DOFs) as defined in Rodgers (2000). I think this should be an easy calculation.

Thank you for the question. The Umkehr method has between 3 and 4 degrees of freedom. It varies slightly with season and latitude of the station.

**Technical corrections:**

In addition to the comments of Reviewer #1, I would like to add these below.

Authors thank reviewer for finding errors throughout the manuscript

P.7, Line 211: COH acronym not defined

This is difficult to correct as COH is the name of the dataset.

We changed the sentence to the following

"The second record is **the NOAA COHesive (COH)** data set that combines records data from the SBUV/2 and OMPS (NOAA processing, further referred to as OMPS_NOAA) instruments on the many satellites using correlation-based adjustments providing an overall bias adjustment plus an ozone dependent factor (SPARC/IO3C/GAW. 2019)."

P.11, Line 355: "It is" -> "This is"

We made the change.

P.11, Lines 335-337: The sentences may need some revision, so that the non-specialist readers do not think that the original method used a Laser.

Thank you for the suggestion. We removed the sentence. The text now reads

"The measurement of a Dobson slit function is not a simple task. The original method used a model 783 McPherson spectrophotometer to determine he slit functions for Dobson 083 (Komhyr et al, 1993). The method restricted the slit function to the core band-pass and did not provide information about out-of-band light rejection."

P.11, Line 37: „Komhyr and Evns, 2006"  -> Komhyr and Evans, 2006

Done

P.21, Line 76: space missing between the period and "This means that the retrieval is…."

Corrected

P.23, Line 725: "NRL climatological" -> NRL climatology

Corrected

P.35, Figure 1: The caption for c) is quite confusing, especially with the usage of multiple ")". Perhaps this can be simplified to:

"c-d) Standard deviations for the mean biases shown in panels a) and b). OPR is operational, and SLC is standard stray light correction."

Thank you for suggested improvement. We adopted your text.

P.37, Figure 4 Caption: "compare operational" -> "compared with operational"

Thank you , we made a correction in the Figure caption.

P.37, Figure 4 Caption: Is it "13-months running smoothing" or "13-month running average"?

Thank you , we made a correction.

P.37, Figure 5 Caption: "(old 6) a" seems misplaced.

Thank you for noticing the error. We removed the confusing reference.

References need to be checked, e.g. Rodgers (1990, 2000) are missing.

Thank you for letting us know we did not include reference to the Rodgers papers. These references were added, and we checked for other missing references.

**Citation**: https://doi.org/10.5194/amt-2021-203-RC2